# Zmiz1 is a novel regulator of lymphatic endothelial cell gene expression and function

**Rajan K. C.**[1], **Nehal R. Patel**[1], **Anoushka Shenoy**[1], **Joshua P. Scallan**[2], **Mark Y. Chiang**[3], **Maria J. Galazo**[1,4], **Stryder M. Meadows**[1,4]*

**1** Department of Cell and Molecular Biology, Tulane University, New Orleans, LA, United States of America, **2** Molecular Pharmacology and Physiology, Morsani College of Medicine, University of South Florida, Tampa, FL, United States of America, **3** Department of Internal Medicine, Division of Hematology-Oncology, Medical School, University of Michigan, Ann Arbor, MI, United States of America, **4** Tulane Brain Institute, Tulane University, New Orleans, LA, United States of America

\* smeadows@tulane.edu

## Abstract

Zinc Finger MIZ-Type Containing 1 (Zmiz1), also known as ZIMP10 or RAI17, is a transcription cofactor and member of the Protein Inhibitor of Activated STAT (PIAS) family of proteins. Zmiz1 is critical for a variety of biological processes including vascular development. However, its role in the lymphatic vasculature is unknown. In this study, we utilized human dermal lymphatic endothelial cells (HDLECs) and an inducible, lymphatic endothelial cell (LEC)-specific *Zmiz1* knockout mouse model to investigate the role of Zmiz1 in LECs. Transcriptional profiling of *ZMIZ1*-deficient HDLECs revealed downregulation of genes crucial for lymphatic vessel development. Additionally, our findings demonstrated that loss of Zmiz1 results in reduced expression of proliferation and migration genes in HDLECs and reduced proliferation and migration *in vitro*. We also presented evidence that Zmiz1 regulates Prox1 expression *in vitro* and *in vivo* by modulating chromatin accessibility at *Prox1* regulatory regions. Furthermore, we observed that loss of *Zmiz1* in mesenteric lymphatic vessels significantly reduced valve density. Collectively, our results highlight a novel role of Zmiz1 in LECs and as a transcriptional regulator of Prox1, shedding light on a previously unknown regulatory factor in lymphatic vascular biology.

## Introduction

Lymphatic vasculature plays a crucial role in maintaining tissue fluid homeostasis, facilitating lipid absorption, and conducting immune surveillance. The proper establishment of a functional lymphatic system is vital for embryonic and postnatal development and normal function. Conversely, aberrant lymphangiogenesis is associated with various diseases, including lymphedema, diabetes, cancer, inflammation, and neurological disorders such as Alzheimer's disease [1]. Extensive research over the years has identified several transcription factors (TFs) that regulate lymphatic development, such as PROX1, GATA2, NFATC1 and FOXC2, thereby shedding light on the role of the lymphatic vasculature in health and disease. Transcriptional co-regulators, acting as activators or repressors, play a crucial role in orchestrating selective

**Data Availability Statement:** All relevant data and Supporting Information are within the manuscript and are available in Gene Expression Omnibus

(GEO) database under the following accession numbers: GSE225057 and GSE225130.

**Funding:** This work was supported by Tulane University start-up funds (SMM), NIH-R01 HL139713 (SMM), NIH-R01 HL163196 (SMM), NIH-R01 NS128106 (MPG), NIH-R01 HL142905 (JPS), NIH-R01 AI136941(MYC) and the Priddy Spark Fund -Tulane University (SMM and MJG). The funders had no role in study design, data collection and analysis, decision to publish, or preparation of the manuscript.

**Competing interests:** The authors have declared that no competing interests exist.

gene transcription in a cell type-specific manner. Alterations in the interaction between TFs and co-regulators can lead to pathological conditions [2,3]. Thus, it is imperative to identify and comprehend the novel regulatory mechanisms involving transcription co-regulators during both physiological and pathological lymphangiogenesis.

Zinc Finger MIZ-Type Containing 1 (Zmiz1) is a member of the PIAS family of proteins and exerts its function as a transcriptional co-activator of Notch [4], Androgen Receptor (AR) [5], p53 [6], Estrogen Receptor [7], and Smad3/4 [8]. Zmiz1 function is critical in diverse developmental processes, such as angiogenesis, as demonstrated by the occurrence of vascular defects and embryonic lethality upon global deletion of Zmiz1 [9]. Specifically, global deletion of Zmiz1 resulted in embryonic lethality at E10.5 due to vascular defects such as an underdeveloped heart valve and atrium and defective yolk sac vasculature in the Zmiz1 knockout mice. Moreover, research has unveiled previously unknown roles for Zmiz1 in multiple diseases, including leukemia [4,10], erythropoiesis [11], osteosarcoma [12], diabetes [13], multiple sclerosis [14] and in a range of neurodevelopmental disorders [15–17]. Notably, Zmiz1 has been identified as a regulator of Notch-dependent T-cell development and leukemogenesis [4,18]. In recent years, the importance of Zmiz1 in the pathogenicity of diabetes and cancer has gained significant recognition as well [19]. However, despite our growing understanding of the many physiological and pathological roles of Zmiz1, there is a substantial knowledge gap concerning its functional and mechanistic roles as a transcriptional co-regulator in the development of lymphatic vasculature and lymphatic endothelial cell (LEC) biology. A deeper understanding of Zmiz1 contributions to the lymphatic vasculature will provide valuable molecular regulatory insights that could also influence the development of novel therapeutic interventions targeting lymphatic vascular defects.

In this study, we investigated the role of Zmiz1 in the regulation of lymphatic development and function. Our results demonstrated that Zmiz1 is robustly expressed in various subtypes of LECs. Silencing Zmiz1 in cultured LECs lead to a downregulation of genes essential for lymphatic vessel development and cell migration and proliferation. Depletion of ZMIZ1 in Human Dermal Lymphatic Endothelial Cells (HDLECs) resulted in impairment of LEC migration and proliferation *in vitro*. Furthermore, genetic ablation of *Zmiz1* in LECs resulted in decreased PROX1 expression and a reduction in the number of mesenteric lymphatic valves. Notably, our data suggested that Zmiz1 regulates Prox1 expression via modulation of chromatin accessibility, revealing an unidentified role for Zmiz1 in LECs. These findings indicated that Zmiz1 has a dynamic role in LECs by functioning as a transcriptional activator and chromatin remodeler to control various cellular processes.

## Methods and materials

### Mouse and treatment

All animal experiments were performed in accordance with Tulane University Institutional Animal Care and Use Committee (IACUC) policies. *Zmiz1^{f/f}* [4], *Prox1-Cre^{ERT2}* [20] and *Ai14* reporter [21] (Stock No. 007914; Jackson Laboratory) mice were crossed to generate tamoxifen inducible, lymphatic endothelial cell-specific Zmiz1 knockout mice (*Zmiz1^{f/f};Prox1-Cre^{ERT2}; Ai14*). For postnatal Cre recombination, newborn offspring were administered Tamoxifen (MilliporeSigma, T5648) orally at a concentration of 100 μg on P1–P3 and mesentery lymphatic vessels were analyzed at P8. For embryonic studies, timed mating was carried out, designating E0.5 on the day a vaginal plug was observed. Mothers were administered 100 ug of tamoxifen via intraperitoneal injection at day E10.5 to induce Zmiz1 deletion in the embryonic LEC lineage.

## Cell culture and RNA interference

Human Dermal Lymphatic Endothelial Cells (HDLECs) (Promocell, C-12216) were cultured in endothelial cell growth medium MV2 (EGM-MV2) (Promocell, C-22022) supplemented with Growth Medium MV 2 SupplementMix (Promocell, C-39226) at 37˚C and 5% $CO_2$. All cell culture experiments were carried out within the 5 passages. HDLECs were transfected with a pool of control non-targeting siRNA (Horizon, D-001810-10-05) or *ZMIZ1* siRNA (Horizon, L-007034-00-0005; this SMARTpool reagent consists of 4 siRNAs each targeting different regions of the Zmiz1 mRNA) for 48 hours using Lipofectamine 3000 (Thermo Fisher, L3000015). The final concentration of siRNA solution was 200 nM. In addition, 2 of the 4 *ZMIZ1* siRNAs were transfected independently to assess for potential off-target effects. Both *ZMIZ1* siRNAs resulted in similar gene expression changes associated with the cocktail *ZMIZ1* siRNA indicating specificity of the designed *ZMIZ1* siRNAs (S5C and S5D Fig).

## Scratch wound healing assay

Scratch wound healing assay was performed as previously described [22]. Briefly, a scratch was made using P200 pipette tip on confluent control and Zmiz1 knockdown HDLECs. Phase contrast images were taken at time 0 hours, 24 hours, and 48 hours using Leica DMi8. Percentage of wound closure and rate of migration were analyzed using ImageJ software.

## Immunocytochemistry

Briefly, cells were fixed, permeabilized, and blocked. Primary antibodies were incubated at 4˚C overnight followed by secondary antibody at room temperature for 1 hour. Fluorescence images were taken using Leica DMi8. Primary antibodies used include Ki67 (Cell Signaling Technology, 9449, 1:100) and Zmiz1 (Cell Signaling Technology, 89500S). Nuclei were stained with NucBlue (Invitrogen, R37606).

## Quantitative PCR (qPCR)

Total RNA from control and ZMIZ1 knockdown HDLECs was extracted using the GeneJET RNA Purification Kit (Thermo Fisher Scientific, K0732). To determine the mRNA expression levels, 1 μg of extracted RNA was transcribed into cDNA using the iScript Reverse Transcription Supermix (Bio-Rad, 1708840). qPCR was performed using the PerfeCTa SYBR Green SuperMix (Quantabio, 95071) on CFX96 system (Bio-Rad). The list of qPCR primers used in this study is included in S1 Table. Relative gene expression was determined using the ΔΔCt method. Three independent biological replicates were used, and three technical replicates were performed per sample.

## Western blot

Total Protein was isolated using the RIPA lysis buffer (Thermo Fisher Scientific, 89901) supplemented with protease inhibitor cocktail (Thermo Fisher Scientific, 78430) and phosSTOP (MilliporeSigma, 04906845001). Protein concentrations were determined using Qubit fluorometer 3.0 (Invitrogen, Q33216) and qubit protein assay (Thermo Fisher Scientific, Q33211). Protein samples were run on 4–20% Mini-PROTEAN TGX Precast gels (Bio-Rad, 4568094) and transferred onto 0.2 μm PVDF membrane (Bio-Rad, 1704156) using Trans-Blot turbo transfer system (Bio-Rad, 1704150). Membranes were blocked for 1 hour with 5% BSA in TBST (0.1% Tween-20 in 1x TBS) and incubated in primary antibodies at 4˚C overnight with agitation. Membranes were washed in TBST three times five minutes each and incubated with secondary antibodies for 1 hour at room temperature with agitation. Following washes, target

proteins were detected using the LI-COR Odyssey imaging system. Band densitometry was quantified using the ImageJ software. The following primary antibodies were used for the Western blot analysis: PROX1 antibody (Abcam, ab199359, 1:1000), Ki67 (Cell Signaling Technology, 11882S, 1:1000) and β-ACTIN (Cell Signaling Technology, 3700S, 1:5,000).

## Immunohistochemistry analysis of mouse mesentery

Immunofluorescence on mouse mesentery was performed as previously described [23]. Briefly, mesentery was dissected, pinned down in sylgard, and fixed in 4% PFA for 6 hours at room temperature. Mesentery was washed three times in 1× PBS and permeabilized in 0.5% TritonX-100/PBS (PBST) at room temperature for 30 minutes, then blocked with CAS-block (Thermo Fisher Scientific, 88120) at 4˚C overnight. Primary antibodies were diluted in CAS-block and incubated with primaries at 4˚C overnight, followed by incubation in appropriate secondary antibodies for 4 hours at room temperature. Mesenteries were washed 3 times in 1× PBS and mounted on a slide using the ProLong Diamond Antifade Mountant (Thermo Fisher Scientific, P36961). Images were taken with a Nikon C2 confocal. The primary antibodies used were anti-CD31 antibody (BD Pharmingen, 553370, 1:500) and anti-PROX1 antibody (Abcam, ab101851, 1:500).

## Lymphangiography

This assay was performed as previously described [24]. P8 pups were fed 1 μl 4,4-Difluoro-5,7-Dimethyl-4-Bora-3a,4a-Diaza-s-Indacene-3-Hexadecanoic Acid BODIPY™ FL C16 (BODIPY FL $C_{16}$, ThermoFisher Scientific, D3821) diluted in olive oil at a concentration of 10 μg/μL. Three hours later, mice were euthanized, and the mesenteric lymphatic vasculature was dissected, mounted, and then imaged using Leica stereomicroscope.

## RNA sequencing and gene expression analysis

RNA extraction and processing for sequencing was performed as previously described [22]. Briefly, total RNA was extracted from control and *ZMIZ1* siRNA transfected HDLECs using GeneJET RNA Purification Kit (Thermo Fisher Scientific, K0732). RNA concentration and RNA integrity (RIN) number were determined using Qubit RNA High Sensitivity Assay Kit (Thermo Fisher Scientific, Q32852) and Bioanalyzer RNA 6000 Nano assay kit (Agilent, 5067–1511) respectively. RNA library was prepared using TruSeq RNA Library Prep Kit v2 (Illumina, RS-122-2001). The sequencing library was quantified and verified using Qubit dsDNA High Sensitivity Assay Kit (Thermo Fisher Scientific, Q32851) and DNA1000 assay kit (Agilent, 5067–1505) respectively. Verified libraries were sequenced using the Nextseq 500/550 High Output kit v2.5 (75 Cycles) (Illumina, 20024906) on a Nextseq 550 system. Sequenced reads were aligned to the human (hg19) reference genome with RNA-Seq alignment tool (STAR aligner). mRNA expression quantification and differentially expressed genes determination was performed using the RNA-Seq Differential Expression tool (version 1.0.1). Both alignment and differential expression analysis were performed on the Illumina BaseSpace Sequence Hub. Overrepresentation analysis of differentially expressed genes was performed using WEB-based Gene SeT AnaLysis Toolkit (WebGestalt) [25]. Sequencing data have been deposited in the Gene Expression Omnibus (GEO) database with accession no. **GSE225057**.

## ATAC sequencing and peak analysis

The ATAC sequencing library was prepared as per manufacturer instruction (Active Motif, 53150). Intact nuclei were isolated from control and *ZMIZ1* siRNA transfected HDLECs.

Samples were treated with a hyperactive Tn5 transposase which tag the target DNA with sequencing adapters and fragment the DNA simultaneously. The library was then quantified using Qubit dsDNA High Sensitivity Assay Kit (Thermo Fisher Scientific, Q32851) and verified using the Bioanalyzer DNA High Sensitivity Assay Kit (Agilent, 5067–4626). Validated samples were sequenced using the NextSeq1000/2000 P2 Reagents (100 Cycles) v3 (Illumina, 20046811) on a Nextseq1000/2000. Resulting sequencing data were analyzed using basepairtech ATAC-Seq pipeline (www.basepairtech.com). Briefly, sequenced reads were aligned to the human (hg19) reference genome using Bowtie2. ATAC-Seq peaks and differentially accessible regions were quantified using MACS2 and DESeq2. Sequencing data have been deposited in the Gene Expression Omnibus (GEO) database with accession no. **GSE225130**.

## Luciferase reporter assay

The ATAC peaks sequence (*Prox1* peak 1 and peak 2) were cloned into pGL4.20-Firefly luciferase (pGL4.20[luc2/puro]; Promega) using HindIII restriction site to generate pGL4.20-ChIPPeak-luc. Cloning was verified using sequencing. pGL4.75[hRluc/CMV] vector (Renilla reniformis) was used as control. HEK-293T cells were transfected with non-targeting siRNA (Horizon, D-001810-10-05) and *ZMIZ1* siRNA (Horizon, L-007034-00-0005) using Lipofectamine3000 (Thermo Fisher, L3000015) (final concentration:200 nM). 24 hours later, cells were transfected with pGL4.20-ATACPeak-luc and pGL4.75. Luciferase activities were assayed using Dual-Luciferase Reporter Assay System (Promega). The expression of firefly luciferase was normalized with renilla luciferase and percentage luciferase activity was calculated relative to the baseline activity of control siRNA treated pGL4.20-ATACPeak -luc.

Peak 1: AGTACAGGCAGCTCAGGCCCAGCTGCCCCAGATAAGAGGTGGCCCGTGTTAATG CACAGGCTTCCTCTGCACCTCAGCAGGGCCTTCCTTTTCTAAACAGTCTCCCTTTAA TGTTG

Peak 2: GCCGGGTACGTCAGATAGACTGTGACGTGCAGTCTTCCTGTTTCCTTCAGC TGTG

TCTTAAAGTAAATCTTGTTGTGGAGCGGAGCCCTCAGCTGAGGGAGCGCTCTGAAATA ATACACCATTG

## Statistical analysis

Data analysis was performed using GraphPad Prism version 9.0.0 for Windows (www.graphpad.com). Data are presented as mean ± SEM. Unpaired 2-tailed Student's *t* tests were used to determine statistical significance assuming equal variance. $P < 0.05$ is considered statistically significant. All experiments were performed with at least 3 biological replicates for each control and experimental group. Illustrations were created with BioRender.com.

## Results

### Zmiz1 is expressed in lymphatic endothelial cells

To assess whether Zmiz1 is expressed in LECs, we first performed ZMIZ1 immunofluorescent antibody staining on cultured human dermal lymphatic endothelial cells (HDLECs) and observed robust nuclear expression of ZMIZ1 (Fig 1A), similar to many other cell types (www.proteinatlas.org, www.uniprot.org, [7]). We next examined Zmiz1 expression in LECs using three publicly available single cell sequencing datasets. Firstly, data from http://betsholtzlab.org/VascularSingleCells/database.html showed robust expression of *Zmiz1* in LECs associated with murine lung ECs [26,27] (Fig 1B). Secondly, utilizing a single cell atlas of adult mouse mesenteric LECs [28], we observed *Zmiz1* expression in collecting, pre-collecting, capillary,

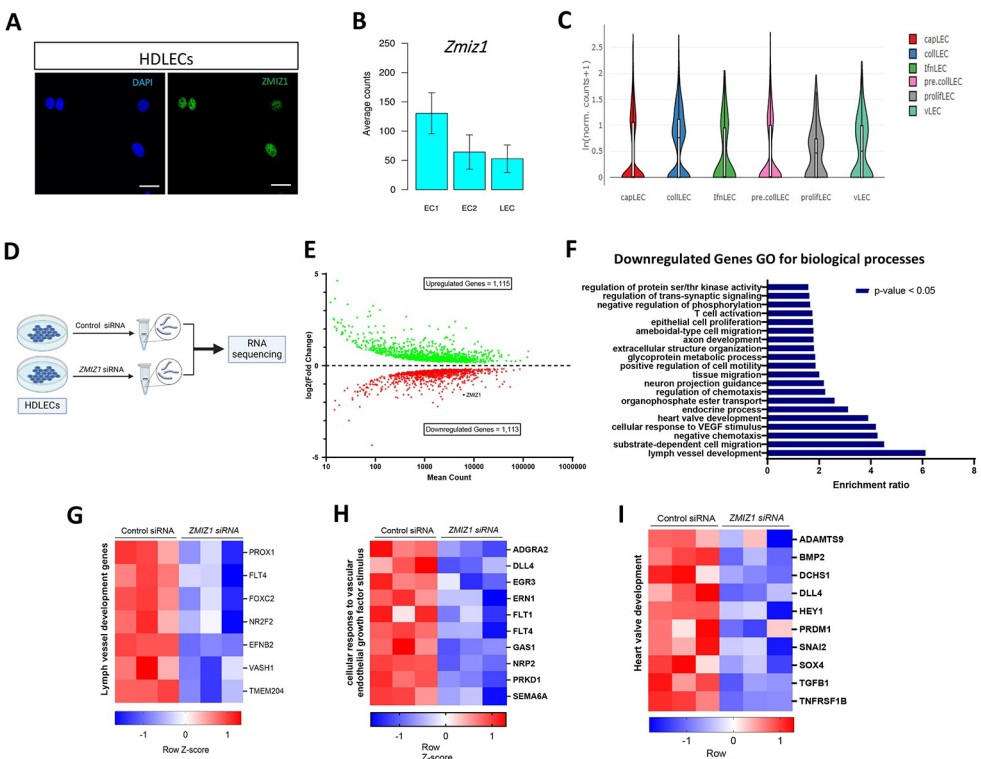

**Fig 1. Loss of Zmiz1 in LECs impairs the expression of lymph vessel development genes.** (A) Immunofluorescence staining for ZMIZ1 (green) and DAPI (blue) showing nuclear localization of ZMIZ1 in HDLECs. Scale bar: 10 μm. (B) *Zmiz1* mRNA expression in ECs and LEC from mouse lungs (adapted from single cell data on http://betsholtzlab.org/ VascularSingleCells/database.html). EC1 and EC2 are different populations of cell clusters that are distinct from artery, vein and capillary ECs. (C) *Zmiz1* mRNA expression in various LEC subtypes (adapted from single cell RNA sequencing dataset of adult mouse mesenteric lymphatic endothelial cells [28]). Capillary (capLECs), collecting (collLECs), precollecting and collecting (pre/collLECs), valve (vLECs), proliferative (prolifLECs), IFN (IfnLECs), endothelial cells (EC) and lymphatic endothelial cells (LEC). (D) Outline of the workflow used to knockdown (KD) *ZMIZ1* in HDLECs for RNA sequencing of control and *ZMIZ1* siRNA treated HDLECs (n = 3). (E) MA plot of differentially expressed genes (DEGs) between control and *ZMIZ1* siRNA HDLECs. (F) Top Gene Ontology (GO) biological process terms enriched in downregulated genes in *ZMIZ1* siRNA (FDR ≤ 0.05). (G-I) Representative clustered heatmaps of gene count Z scores for lymph vessel development genes (G), cellular response to vascular endothelial growth factor stimulus (H), and heart valve development genes (I).

and valve LECs (Fig 1C). Finally, we found varying levels of *Zmiz1* expression in different LEC subtypes surveyed via a single-cell transcriptional roadmap of the mouse and human lymph nodes [29] (S1 Fig). Overall, these expression datasets show that Zmiz1 is expressed in several subtypes of LECs, thereby establishing its presence in the lymphatic endothelium and indicating a potential role for Zmiz1 in lymphatic vessel development.

## Loss of Zmiz1 alters lymphatic gene expression profiles, including developmental genes

The role of Zmiz1 in the LECs is unknown. To begin to assess Zmiz1 contributions to LEC biology, we knocked down *ZMIZ1* in HDLECs using siRNA and performed RNA sequencing (RNA-Seq) (Fig 1D and S2). RNA-Seq analysis revealed 2,228 differentially expressed genes (DEGs) (1,115 upregulated and 1,113 downregulated genes) upon *ZMIZ1* knockdown (Fig 1E). Gene ontology (GO) analysis for biological processes showed that downregulated genes were enriched in numerous processes, including lymph vessel development, heart valve

development, and cellular response to vascular endothelial growth factor stimulus (Fig 1F). Heat map signatures of the genes associated with these processes highlighted various factors with prominent and often multiple functions in LECs (Fig 1G–1I). For instance, Vash1 regulates secondary sprouting and EC proliferation and its depletion leads to defective trunk lymphatic vessel formation in the zebrafish trunk [30]. TMEM204, also known as claudin like protein 24, is known to interact with VEGFR2 and 3 and regulates lymphatic vessel patterning and development [31]. Among the lymph vessel development-related genes, PROX1 is a master regulator of lymphatic development and has been associated with many lymphatic defects, along with FLT4 and FOXC2 [1]. Further GO analysis of the upregulated DEGs implicated Zmiz1 in additional biological processes, such as cell adhesion and response to wound healing, while KEGG pathway analysis of all DEGs linked various signaling and process-related pathways to these various biological processes (S3 Fig). Taken together, the RNA-seq results indicated Zmiz1 transcriptionally regulated LEC gene expression and may have broad impacts on lymphatic vascular development and function.

## Impaired cell migration and proliferation in Zmiz1 deficient LECs

To further define the main processes governed by Zmiz1 function, enrichment network assessment of the RNA-seq derived GO terms was performed. This analysis pointed towards a critical role for Zmiz1 in LEC migration, lymphatic vessel development and cell adhesion (Fig 2A). Further examination revealed downregulation of DEGs tied to both cell migration and the cell leading edge, as depicted via heat maps (Fig 2B and 2C). Cell migration is a complex process that requires remodeling of the actin cytoskeleton, cell-matrix adhesion and interpretation of guidance signals at the cell leading edge or tip cells [32] in the endothelium—processes detected in our GO analyses. Therefore, to investigate the role of Zmiz1 in migration, we performed *in vitro* scratch wound healing assays with HDLECs treated with either control or *ZMIZ1* siRNA. Compared to control siRNA LECs, *ZMIZ1* siRNA treated cells exhibited a significant reduction in wound closure and cell migration rate after 24 hours (Fig 2F and 2G).

Although our results indicated defective cell migration in the absence of Zmiz1, there are other factors that could account for this observation. Decreases in cell proliferation could also lead to similar wound healing results. Accordingly, GO analysis of Zmiz1 downregulated genes implicated cell proliferation changes (Figs 1F and 2D). Subsequently, we performed a scratch assay and used Ki67 as a proliferation marker to investigate the effect of Zmiz1 on LEC proliferation. We observed a significant decrease in Ki67 positive cells in both scratch and non-scratch regions upon Zmiz1 knockdown in HDLECs (Fig 2H–2J). Therefore, to determine whether Zmiz1 could affect cell migration, independent of changes to proliferation, HDLECs were treated with the DNA synthesis inhibitor cytosine β-D-arabinofuranoside; a similar number of Ki67 positive cells were detected in both control and *ZMIZ1* siRNA treated HDLECs (S4 Fig). Examination of wound healing under the same conditions demonstrated that HDLECs treated with *ZMIZ1* siRNA displayed significant reductions in wound closure and cell migration rates when compared to control cells (Fig 2K–2M). Overall, these findings showed that loss of Zmiz1 negatively affects LEC proliferation and migration and are consistent with previous Zmiz1 studies in T-cells [4,18], melanocytes [33] and osteosarcoma [12].

## Zmiz1 modulates chromatin accessibility in LECs

Previous studies suggested that Zmiz1 also functions as a chromatin remodeler through the SWI/SNF complex [5]. To assess if Zmiz1 regulates chromatin accessibility in LECs, we performed Assay for Transposase-Accessible Chromatin with high-throughput sequencing (ATAC-Seq) on control and *ZMIZ1* siRNA treated HDLECs (Fig 3A). Initial ATAC-seq

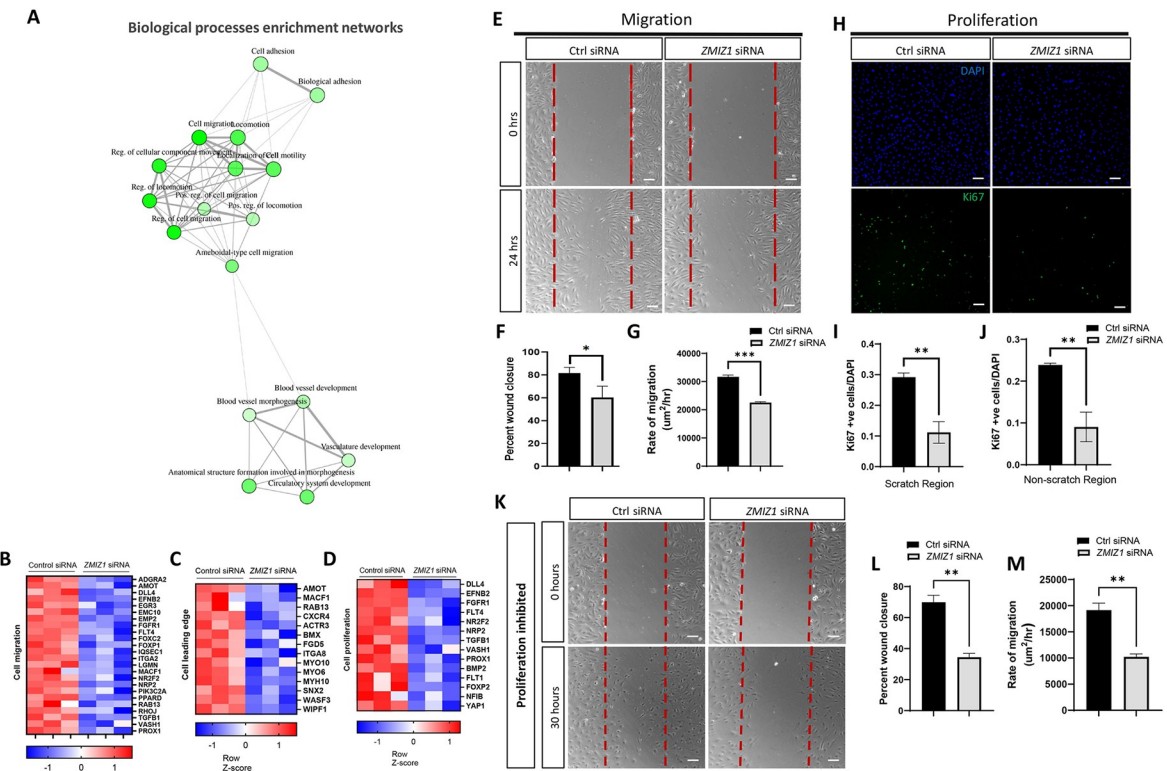

**Fig 2. Reduced LEC migration and proliferation in Zmiz1 deficient LECs.** (A) Enrichment network of biological processes obtained from DEGs in *ZMIZ1* siRNA treated HDLECs. The main processes identified center on development, migration, and adhesion. (B-D) Representative clustered heatmaps of gene count Z scores for cell migration genes (B), cell leading edge genes (C) and cell proliferation genes (D). (E) Scratch wound assays performed on confluent HDLECs following control and *ZMIZ1* siRNA treatment. Imaged at 0 and 24 hours following the scratch. Scale bars: 100 μm. (F-G) Quantification of the percent of wound closure (F) and rate of migration (G) in control and *ZMIZ1* siRNA treated HDLECs at 24 hours post scratch. (H) Cell proliferation marker Ki67 (green) immunostaining on control and *ZMIZ1* siRNA treated HDLECs. DAPI (blue), scale bars: 100 μm. (I-J) Quantification of Ki67 positive (+ve) cells/DAPI in 10X objective field in scratch region (I) and non-scratch region (J) of control and *ZMIZ1* siRNA treated HDLECs at 24 hours post scratch. (K) Scratch wound assays performed on control siRNA and *ZMIZ1* siRNA treated, cytosine β-D-arabinofuranoside proliferation controlled confluent HDLECs at 0- and 30-hours post scratch. (L-M) Quantification of the percent of wound closure (L) and rate of migration (M) in treated control and *ZMIZ1* siRNA HDLECs at 24 hours post scratch. *P < 0.05, **P <0.01, ***P <0.001 calculated by unpaired Student's t-test. All experiments were performed in triplicate and repeated three times.

analysis of the combined data sets showed that the majority of sequencing peaks were distributed in intergenic and intronic regions, while few were seen in promotor and exon regions (Fig 3B). However, in the peak subset corresponding to promoters, a reduction in signal intensity near the Transcription Start Site (TSS) was observed in *ZMIZ1* siRNA treated LECs compared to control cells (Fig 3C). Analysis of differential accessibility regions (DARs) identified 3639 and 1808 genes with reduced or increased chromatin accessibility, respectively, in ZMIZ1 deficient HDLECs (Fig 3D). In addition, the top enriched motifs pulled from DARs included HOX2A, GATA3, CUP9, ZEB1, FOXO1, SMAD3, and AR (Fig 3E) and implicated these factors as lymphatic Zmiz1 co-regulators. Integration of the RNA-seq and ATAC-seq datasets revealed genes likely to be influenced directly by Zmiz1: evaluation of downregulated DEGs and downDARs (decreased accessibility) showed 170 overlapping genes, while only 16 genes with overlapping upregulated DEGs and upDARs (increased accessibility) were identified (Fig 3F), suggesting a transcriptional activating role for Zmiz1. GO analysis of these overlapping genes indicated enrichment in several biological processes, with highest enrichment associated with lymph vessel development (Fig 3G).

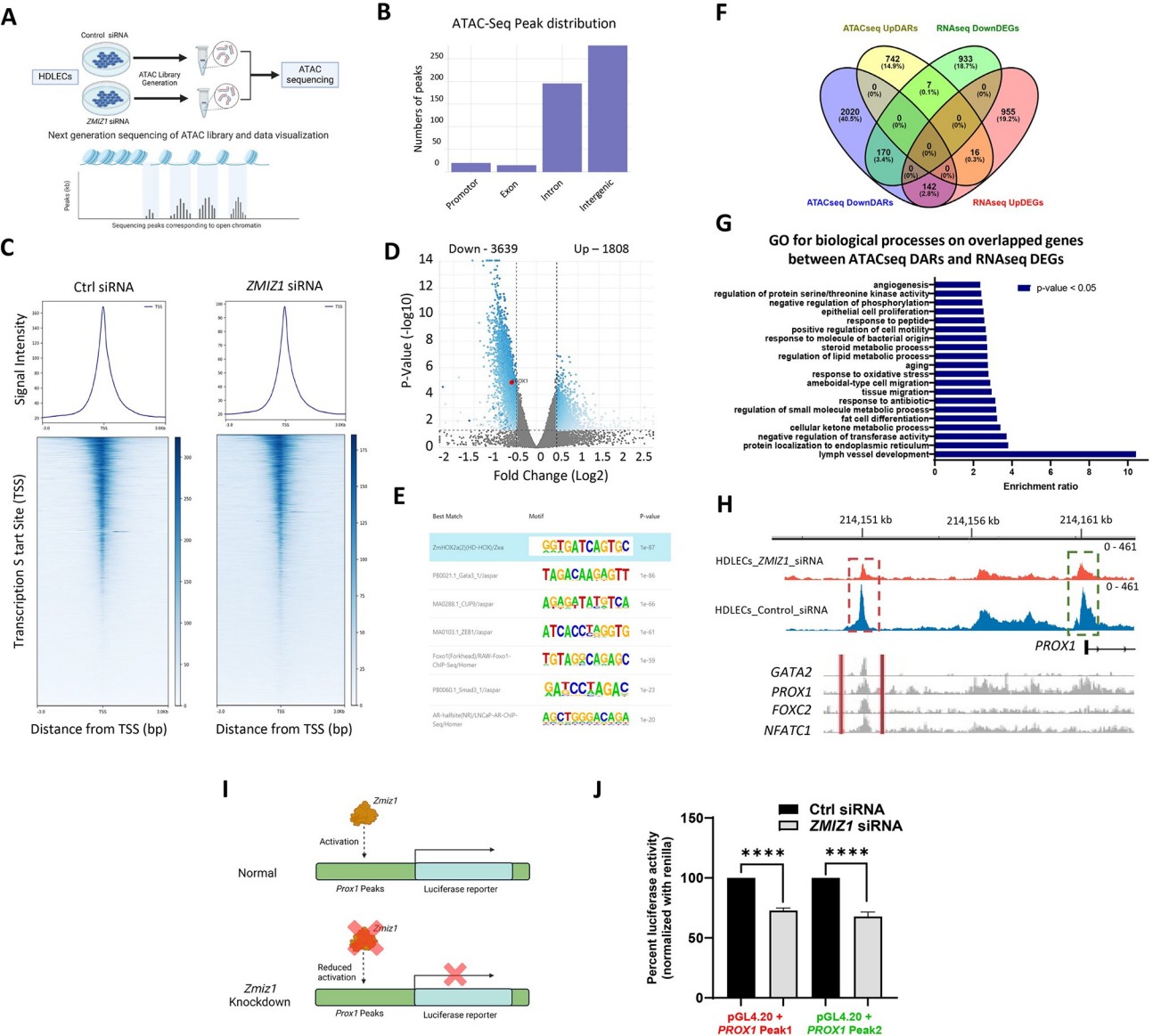

**Fig 3. Zmiz1 facilitates chromatin accessibility in LECs.** (A) ATAC sequencing experimental paradigm on HDLECs treated with control and *ZMIZ1* siRNA. (B) Genomic distribution of ATAC-seq peaks. (C) Heatmap depicting the ATAC-seq profile around protein coding transcription start sites (TSS). (D) Volcano plot of differentially accessible regions (DAR) between *ZMIZ1* and control siRNA HDLECs. *PROX1* is highlighted in red. (E) Top enriched motif sequences. (F) Venn diagram depicting gene overlap among ATAC-seq downDARs and upDARs, and RNA-seq down and upregulated DEGs. (G) GO for biological process in overlapped genes between ATAC-seq DARs and RNA-seq DEGs. (H) Representative ATAC-seq tracks on the human *PROX1* gene. ATAC-seq peaks for *ZMIZ1* (orange) and control (blue) siRNA HDLECs. Red dotted box, Prox1 enhancer; green dotted box, Prox1 promoter. ChIP-seq peaks for GATA2, PROX1, FOXC2, and NFATC1 in the *PROX1* enhancer are depicted in grayscale. I) Schematic of luciferase reporter assay for testing activity of *PROX1* ATAC-seq peaks. J) Percent luciferase activity in control and *ZMIZ1* siRNA treated HEK-293T cells showing reduction of luciferase upon loss of Zmiz1 (Peak 1, red dotted box; and Peak 2, green dotted box depicted in H).

Notably, in the absence of Zmiz1, Prox1 is one of the factors associated with the lymph vessel developmental GO term (Fig 1G) and identified among the 170 genes that exhibited reduced expression in the RNA-seq data and contained downDARs in the ATAC-seq data (Fig 3D and 3F). Closer analysis of accessible chromatin sites revealed open chromatin localized at intergenic and promoter regions of *PROX1* were significantly less accessible following *Zmiz1* depletion (Fig 3H). The upstream intergenic peak was recently identified as an

evolutionary conserved Prox1 enhancer bound by key lymphatic endothelial transcriptional regulators such as GATA2, PROX1, FOXC2, and NFATC1 [34] (Fig 3H). Furthermore, we observed a reduction in open chromatin accessibility in multiple downregulated lymph vessel development genes upon Zmiz1 knockdown, including FLT4, NR2F2, FOXC2, EFNB2, and VASH1 (S5A Fig); qPCR analysis of ZMIZ1 siRNA treated HDLECS further confirmed decreased expression of several important lymphatic development genes, such as *EFNB2*, *COUPTFII*, *FLT4* and *FOXC2* (S5B–S5D Fig). In addition, we performed luciferase reporter assays to validate the regulation of *PROX1* by ZMIZ1 via these two specific DNA regions (Fig 3I). Accordingly, we observed significant decreases in luciferase expression in *ZMIZ1* siRNA treated HEK-293T cells compared to control siRNA treated cells (Fig 3J). Control experiments verified ZMIZ1 presence in the nucleus of HEK-293T cells and that in comparison to control siRNA, ZMIZ1 siRNA led to significant reduction in expression of both *ZMIZ1* and *PROX1* mRNA (S6 Fig). Overall, the ATAC-seq findings indicated Zmiz1 regulates gene expression by binding to genomic regulatory regions, largely residing in areas typical of enhancers, and influencing chromatin accessibility.

## Zmiz1 regulates Prox1 expression *in vitro* and *in vivo*

Because Prox1 is crucial for proper LEC specification and lymphatic vessel development [35], it was important to confirm the validity of our RNA-seq data showing decreased *PROX1* in *ZMIZ1* depleted HDLECs (Fig 1G). Using siRNA mediated knockdown, western blot (Fig 4A and 4B) and quantitative PCR (qPCR) (Fig 4C) analyses confirmed significant reduction of Prox1 protein and mRNA, respectively, in *ZMIZ1* siRNA treated cells compared to controls. Moreover, we analyzed apoptosis in the same *in vitro* settings and validated that the reduction in Prox1 expression was not associated with an increase in cell death of LECs subjected to *ZMIZ1* siRNA (S7A and S7B Fig).

To corroborate Prox1 downregulation upon loss of Zmiz1 *in vivo*, we generated inducible, lymphatic EC-specific conditional *Zmiz1* knockout mice by breeding *Zmiz1*-floxed [4] and *Prox1-Cre^ERT2* [20] mouse lines. *Prox1-Cre^ERT2;Zmiz1^f/f* mice undergoing gene deletion were referred to as *Zmiz1-KO* mice, while *Zmiz1^f/f* mice lacking *Prox1-Cre^ERT2* were used as control mice. The lymphatic vasculature was further visualized by incorporating the *Rosa26*-Ai14 reporter [21]. Cre-mediated recombination of *Zmiz1* was induced by administrating Tamoxifen orally at Postnatal day (P)1, 2, and 3 and investigating Prox1 expression in the lymphatic vessels of the mesentery at P8 (experimental paradigm illustrated in Fig 4D). Immunofluorescent antibody staining experiments demonstrated a significant reduction in PROX1 levels in *Zmiz1-KO* mice compared to *Zmiz1^f/f* mice that were readily visible in mesenteric LECs and as quantified by fluorescent intensity (Fig 4E and 4F). Thus, similar to the *in vitro* data, expression of Prox1 is substantially reduced in *Zmiz1* deficient LECs *in vivo*.

## Loss of Zmiz1 leads to reduction in mesenteric valve number

It is known that specified LECs in valve forming regions are high in PROX1 and that PROX1 dosage is critical for the formation of lymphovenous valves [1,36]. Investigating further, we analyzed the morphological structure of mesenteric lymphatic vessels and valves at P8 and P20. In *Zmiz1*-KO mice, the mesenteric vessels appeared normal but collecting lymphatic vessels had significantly fewer valves per millimeter than controls (Figs 4G, 4H and S8). Next, we tested the forward flow of lymph fluid, which is often impaired upon lymphatic valve defects. We fed fluorescent lipid, BODIPY FL $C_{16}$, which is selectively absorbed by the collecting lymphatic vessels draining the intestine (S9A Fig). There was no obvious difference in florescent lipid uptake after 3 hours between *Zmiz1-KO* and control

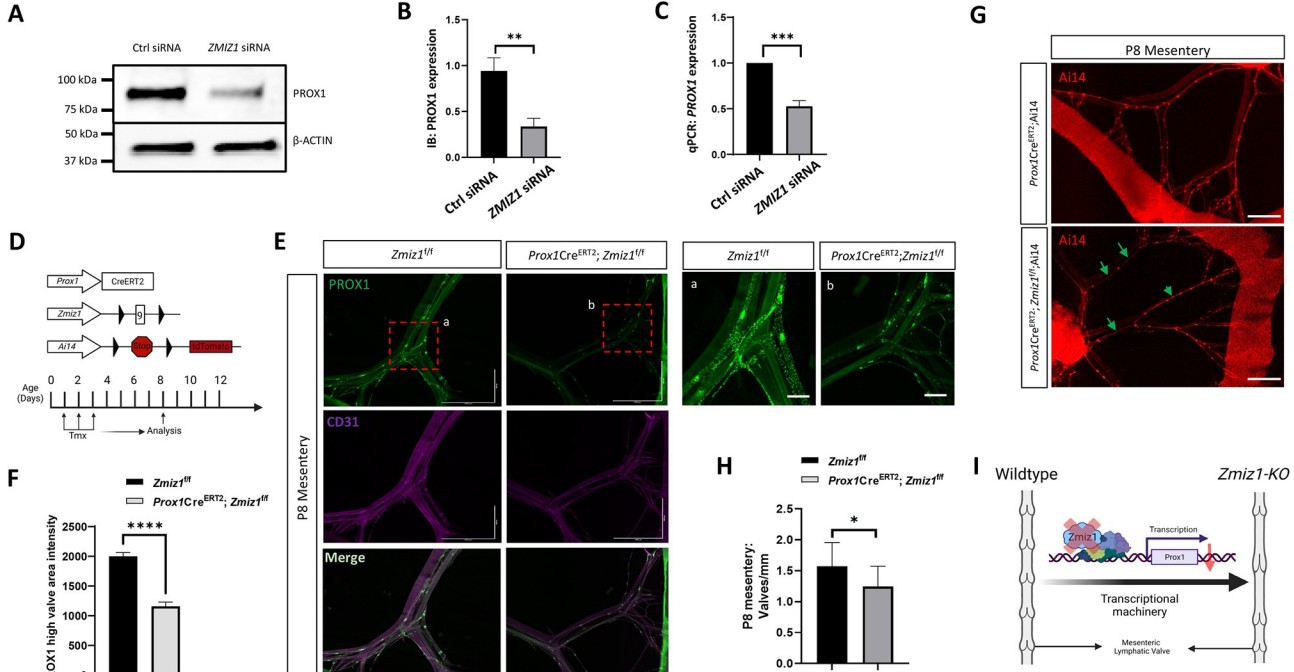

**Fig 4. Zmiz1 regulates Prox1 expression and genetic deletion of *Zmiz1* results in decreased valve number.** (A) Western blot analysis for PROX1 on control and *ZMIZ1* siRNA HDLECs. (B) Densitometric quantification of (A). (C) qPCR analysis for PROX1 expression levels in control and *ZMIZ1* siRNA HDLECs. *n* = 3–5 for each experimental sample. (D) Graphical representation of Cre-LoxP system components and tamoxifen injection procedure for postnatal deletion of *Zmiz1*. (E) Wholemount immunostaining of PROX1 (green) and CD31 (magenta) in *Zmiz1* wild type and mutant postnatal day (P) 8 mesentery indicating reduction in PROX1 expression. Scale bars: 1000 μm (B), 200 μm (a, b). (F) PROX1 high valve area intensity quantification in wild type and *Zmiz1*-KO mesentery. (G) Fluorescence imaging of the morphology of the P8 mesenteric lymphatic vasculature indicated by RFP expression. No significant differences in lymphatic vessel diameters were observed between Zmiz1 wild type and mutant mice. Abnormal RFP positive valve arrangements are represented by green arrows. Scale bars: 1000 μm. (H) Valves per millimeter from wild type and *Zmiz1*-KO P8 mesentery. (I) A schematic representation of ZMIZ1 regulation on PROX1 and lymphatic mesenteric valve organization. At least three controls and 3 knockout mesenteries were used in each analysis, n = 3–5. All values are mean ± SEM. **P <0.01, ***P < 0.001, ****P < 0.0001 calculated by unpaired Student's t test.

mesenteric collecting lymphatic vessels (S9B Fig) suggesting that these vessels have normal forward lymph flow. In addition to the mesentery, we examined the lymphatic vessels of the ears via VEGFR3 immunofluorescent antibody staining; no notable differences in lymphatic density or valve number, arrangement or structure between control and mutant P8 neonates were identified (S10 Fig).

Interestingly, our RNA-seq data indicated reduced expression of genes associated with the GO term for heart valve development (Fig 1I) and reduced valve formation has been linked to lymphatic defects like edema [37,38]. To determine if *Zmiz1* deletion leads to lymphatic defects such as edema, we performed early deletion of *Zmiz1* at Embryonic day (E) 10.5 when LECs are specified and examined embryos at E14.5 (S11A Fig). There were no gross morphological abnormalities in the overall embryo structure, and no edema or lymph fluid leakage was observed in the abdomen (S11B Fig). Further analysis of the mesentery showed no clear changes in lymphatic vessel morphology or in expression of PROX1 and VEGFR3 (S11C Fig). In summary, we observed significant alterations in postnatal mesenteric lymphatic valve number after *Zmiz1* deletion (graphically summarized in Fig 4I), but this abnormality did not impair lipid uptake and there were no significant embryonic changes in lymphatic valve number or lymphatic-associated defects like edema or, chylous ascite.

## Discussion

The role of Zmiz1 in lymphatic endothelial cells and lymphatic development is unknown [4–6,8,18]. In this study, we utilized an inducible *Prox1-Cre*<sup>ERT2</sup> mouse model to investigate the effects of Zmiz1 loss of function *in vivo*, as well as using siRNA knockdown of *ZMIZ1 in vitro* in HDLECs. Our transcriptional profiling of HDLECs with Zmiz1 knockdown revealed impaired expression of migration, proliferation, and lymphatic vessel development genes, with a significant reduction in the expression of *PROX1*—a master regulator of lymphatic vasculature development (Figs 1 and 2). We also found that Zmiz1 facilitates chromatin accessibility in *PROX1* genomic regulatory regions to regulate its expression. Furthermore, our mouse model showed a significant reduction in mesenteric lymphatic vessel valves upon Zmiz1 ablation (Fig 4). These findings suggest a potential role for Zmiz1 in lymphatic development, warranting further investigation.

Proliferation and migration of LECs are critical for lymphatic vessel growth and remodeling during developmental and pathological conditions, where disruption is implicated in wound healing, inflammation, immune responses, and cancer [39]. Downregulated genes for cell migration and proliferation in LECs include common genes such as *PROX1*, *DLL4*, *FLT1*, *BMP2*, *NR2F2*, *FOXC2*, *TGFβ1*, *FLT4*, and *EFNB2* (Fig 2B–2D). These genes are known to induce molecular changes that stimulate LEC migration and proliferation [40–42]. In addition, downregulated cell leading edge genes modify cytoskeletal dynamic at the tip cells or leading edge cells, which also plays a critical role in cell migration [43]. Selection and migration of endothelial tip cells are mediated through Notch-Vegfr, Slit-Robo, and Semaphorin pathways [44], which were affected upon loss of Zmiz1 (Figs 1D, 1F and S3). Notch1-Dll4 signaling is also critical for tip cell leader migration [45]. On the other hand, cellular proliferation is mediated through factors such as cyclins, which are controlled by different signals, including Notch, STAT3, and by E3 ligases [46,47]. Interestingly, Zmiz1 is a member of Protein Inhibitor of Activated STAT (PIAS) implicated in STAT pathway and it acts as a Notch1 co-activator [18] and E3 SUMO ligase [15], suggesting that it may potentially regulate proliferation through Notch and E3 ligase activity. Thus, our transcriptomic analyses strongly implicated Zmiz1 in cell migration and proliferation regulation, which was further substantiated by the *in vitro* HDLEC studies showing that Zmiz1 knockdown leads to defective cell migration and proliferation activity.

Gene expression profiles can provide insight into potential gene functions within specific cell types. Zmiz1 is highly expressed in different subtypes of LECs, indicating a potential role for Zmiz1 in these cells. RNA-Seq analysis revealed that loss of Zmiz1 in LECs led to downregulation of lymphatic vessel development genes, including *PROX1*, *FLT4*, *FOXC2*, *NR2F2* and *EFNB2*. These genes are major players in LEC specification, maturation, and valve morphogenesis. We further demonstrated that Zmiz1 potentially regulates Prox1 expression both *in vivo* and *in vitro*. Prox1 is critical for the development of the murine lymphatic system, mainly LEC specification and lymphatic valve morphogenesis through the transcriptional network of Sox18/Coup-TFII and Gata2/Foxc2/Nfatc2 [35,48,49]. Prox1 acts in a dosage dependent manner to regulate formation of lymph venous valve and control specialization of lymphatic valve territory and postnatal maintenance [36,50]. Accordingly, we characterized a significant reduction in mesenteric lymphatic vessel valve density in *Zmiz1*-KO mice (Fig 4G and 4H). We also observed a reduction in cardiac valve development genes with unknown roles in lymphatic valve development that may provide future lines of study (Fig 1I), as these genes are important for valvulogenesis and their dysregulation can lead to valve malformation or pathogenesis in the heart [51,52]. Moreover, development of functional cardiac lymphatic vessels requires Prox1 for LEC specification, budding and transmigration [53] and brings about the

possibility of Zmiz1-regulated valvulogenesis in the heart. Subsequently, the reduction in expression of Prox1 and other valve related genes may contribute to the decrease in mesenteric lymphatic vessel valve density observed upon loss of Zmiz1.

Phenotypically, however, *Zmiz1*-KO mice did not display severe lymphatic vessel defects, which could be attributed to Zmiz1 acting as a modulator to maintain gene expression of Prox1 and relevant genes in LECs. Indeed, in support of this notion, we found that PROX1 expression appeared unchanged in the mesenteric lymphatic vessels of E14.5 Zmiz1-KO embryos (S11 Fig). Nevertheless, it was surprising that changes in expression of numerous genes with fundamental roles in lymphatic vessel development do not result in more extensive phenotypes. One possible explanation is that Zmiz2 is compensating in some manner for the loss of Zmiz1; based on our survey of single cell sequencing databases, Zmiz2 is also expressed in the various subtypes of lymphatic ECs. Other potential explanations may derive from our transcriptomic data obtained in HDLECs. For instance, several pro-angiogenic genes, such as transforming growth factor-β (TGF-β) and kruppel-like factor 4 (KLF4) were elevated and may compensate for the Zmiz1-associated defects. TGF-β signaling maintains the structure of lymphatic vessels and lymphatic homeostasis such that TGF-β signaling reduction resulted in abnormal lymphatic vessel structure, lymphatic vessels network, lymphatic drainage, and affects lymph angiogenesis [54,55]. KLF4 is well known for its regulation of angiogenesis. In lymphatics, *Klf4* deletion resulted in defective lymphatic vessels, branching morphogenesis and decreased lymphatic density [56] and is a valve forming gene [57]. Intriguingly, FOXO1 is significantly downregulated in Zmiz1 deficient HDLECs; FOXO1 is a repressor of lymphatic mesenteric valves and *FoxO1* knockout mice exhibited increased mesenteric lymphatic vessels and valve numbers [57,58]. Increased expression of FoxO1 might counterbalance the valve defects induced by loss of Zmiz1. Another possible reason could stem from differences in the *in vivo* and *in vitro* settings. For instance, perhaps the robust gene expression alterations seen in HDLECs are less pronounced and encompassing *in vivo*. A similar transcriptomic evaluation of LECs from *Zmiz1*-KO mice would provide a clearer correlation to the *in vitro* findings and is therefore an important future step to better understand the transcriptional impact of Zmiz1 on the lymphatic vasculature.

Transcription factors and cofactors play a crucial role in regulating gene expression and controlling the transcriptional program essential for cell specification and differentiation. Previous studies identified Zmiz1 as a transcriptional coactivator for several transcription factors, including p53, androgen receptor (AR), Smad3/4, and notch1. Intriguingly, analysis of our ATAC-seq data implicated MOTIFs for AR and SMAD3, suggesting a similar transcriptional regulatory relationship with Zmiz1 in controlling LEC gene expression. Additional MOTIFs, such as HOX2A, GATA3, CUP9, ZEB1 and FOXO1 point to novel Zmiz1 co-regulators, reflecting a need to explore these potential dynamic partnerships in LECs. At another level of gene regulation, chromatin remodelers fine tune the activity of transcriptional regulators by modulating chromatin accessibility. In this study, ATAC-seq analysis revealed a significant loss of open chromatin in a *PROX1* enhancer and its promoter in response to loss of Zmiz1. Given that Zmiz1 has been shown to transiently cooperate with BRG1 (SMARCA4) and BAF57 of the chromatin remodeling complex SWI/SNF [5], and that Zmiz2 has been shown to interact with SWI/SNF [59], it is possible that Zmiz1 acts as a chromatin remodeler to drive expression of Prox1 and other genes. The epigenetic regulator BRG1 is known to play a crucial role in the activation of Coup-TFII and Lyve1 during lymphangiogenesis [35]. Therefore, our findings suggest that Zmiz1 may play a critical role in regulating lymphatic development through its involvement in chromatin remodeling to affect gene expression.

Another perspective of studying Zmiz1 and Prox1 regulation lies in their multifunctionality in various tissues. While Zmiz1 is well studied in relation to Notch pathways and regulation of

T-cell development [4], Prox1 has been extensively studied in its functional and molecular role in lymphatic vasculature development as well as its implication in the differentiation of myoblasts by interacting with NFAT and Notch pathways [60]. Notch and p53 are common pathways distorted in various cancers, and Zmiz1 and Prox1 are both implicated in tumorigenesis. Specifically, Prox1 dysregulation is linked to breast cancer progression and metastasis [61] and glioblastoma invasion [62], while Zmiz1 is associated with breast cancer and prostate cancer [19]. Recent case studies have also revealed Zmiz1 variants as a cause for various syndromic neurodevelopmental disorders [15–17,63] and risk factors for autism spectrum disorder [64]. Neurodevelopmental disorders such as intellectual disability, autism, and learning difficulties are often linked with impairment in hippocampus dependent learning and memory [65,66]. The hippocampus, particularly the dentate gyrus, plays a key role in learning and memory and is an important place for adult neurogenesis. Prox1 is abundantly expressed in dentate gyrus, and it specifies and regulates granule cell identity and maturation [67,68], and is important for morphogenesis and differentiation of the dentate granule neuron subtypes [69]. Therefore, with respect to Prox1 and beyond, examining the role of Zmiz1 in hippocampal neurogenesis and subsequent neural identity specification, connectivity, and pathological association with neurological and psychiatric disorders may provide new insights into this important topic.

In conclusion, this study highlights the novel role of Zmiz1 in regulating lymphatic development genes in LECs, with a focus on the regulation of Prox1. Our findings suggest that Zmiz1 potentially remodels chromatin accessibility at Prox1 genomic loci. This study provides the first evidence, to our knowledge, of an association between Zmiz1 and Prox1, a gene that plays a critical role in establishing a functional lymphatic vascular network. Additionally, our results suggest that Zmiz1 may have a role in valvulogenesis, cell migration and proliferation, and other cellular processes, indicating the need to further delineate the potential multifunctionality of Zmiz1, especially *in vivo*.

## Supporting information

**S1 Fig. Zmiz1 expression comparison in mouse and human lymph node.** (A, C) Single cell UMAP of human (A) and mouse (C) lymph node lymphatic endothelial cells (LEC) subtypes. (B, D) Zmiz1 expression in distinct LEC subtypes in human (B) and mouse (D) lymph node. Adapted from https://cellxgene.cziscience.com/collections/9c8808ce-1138-4dbe-818c-171cff10e650 [29].
(TIF)

**S2 Fig. ZMIZ1 siRNA treated HDLECs exhibit depleted expression of *Zmiz1*.** (A) qPCR analysis and (B) immunofluorescent antibody staining for ZMIZ1 confirms loss of Zmiz1 expression in HDLECs treated with ZMIZ1 siRNA, as compared to control siRNA treatments (n = 3). All values are mean ± SEM. ***P < 0.001 calculated by unpaired Student's t test.
(TIF)

**S3 Fig. Gene ontology (GO) analysis of differentially expressed genes in *ZMIZ1* siRNA HDLECs compared to control.** (A-B) Top biological processes (A) and KEGG pathway (B) enriched in both upregulated and downregulated genes following loss of Zmiz1 in HDLECs. (C) Top biological processes enriched in upregulated genes following loss of Zmiz1 in HDLECs. p-value <0.05.
(TIF)

**S4 Fig. Proliferation inhibition in wound healing assay.** (A) Proliferation inhibition using Cytosine β-D-arabinofuranoside for 3 hours in HDLECs treated with ctrl and *ZMIZ1* siRNA. DAPI (blue), Ki67(green). (B) Quantification for Ki67 positive (+ve) cells/DAPI (10X field)

show no difference in rate of proliferation (n = 3). All values mean ± SEM. Scale bars: 100 μm. ns–not significant, calculated by unpaired Student's t test.
(TIF)

**S5 Fig. Reduced chromatin accessibility in lymph vessel development genes.** (A) ATAC-seq peaks for *FLT4*, *NR2F2*, *FOXC2*, *EFNB2*, and *VASH1* in control and *ZMIZ1* siRNA treated HDLECs. ATAC-seq peaks are colored blue (control (Ctrl) HDLECs) and orange (*ZMIZ1* siRNA HDLECs). (B) qPCR analysis of control and *ZMIZ1* siRNA treated HDLECs confirm reduced expression of lymphatic development genes *EFNB2*, *COUPTFII*, *FLT4* and *FOXC2* in the absence of Zmiz1 (n = 3). (C-D) qPCR analysis of *ZMIZ1*, *PROX1*, *EFNB2*, *FLT4*, *COUPT-FII*, and *FOXC2* using two different individual *Zmiz1* siRNAs from B (n = 3). All values are mean ± SEM. *P <0.05, **P < 0.01, ***P < 0.001, ****P < 0.0001 calculated by unpaired Student's t test.
(TIF)

**S6 Fig. Confirmation of ZMIZ1 expression and downregulation in HEK-293T cells.** (A) HEK-293T cells fluorescently immunolabeled for ZMIZ1 (red) and DAPI (blue) showed nuclear expression of ZMIZ1. Scale bars: 20 μm (B) qPCR analysis of control and *ZMIZ1* siRNA treated HEK-293T cells confirm significant downregulation of both *ZMIZ1* and *PROX1* mRNAs. n = 3. All values are mean ± SEM. ***P < 0.001 calculated by unpaired Student's t test.
(TIF)

**S7 Fig. Loss of Zmiz1 is not associated with apoptosis in LECs.** (A) HDLECs treated with Staurosporine, an inducer of cell death, and stained with the apoptotic marker cleaved caspase 3 (CCASP3) serve as a positive control. (B) HDLECS treated with control and ZMIZ1 siRNAs and immunofluorescently labeled for DAPI (blue), CCASP3 (green) and KI67(green). No apoptotic cells were observed in either treatment. n = 3; scale bars: 50 μm.
(TIF)

**S8 Fig. Genetic deletion of *Zmiz1* results in decreased valve number at P20.** (A) Fluorescence imaging of the morphology of postnatal day (P) 20 mesenteric lymphatic vasculature indicated by RFP expression in *Prox1*CreERT2;Ai14 control and *Prox1*CreERT2;*Zmiz1*^f/f^;Ai14 mutant mice. Abnormal RFP positive valve arrangements are represented by green arrows. No significant differences in lymphatic vessel diameters were observed between Zmiz1 wild type and mutant mice. Scale bars: 1 mm. Three control and Zmiz1 knockout mesenteries were used.
(TIF)

**S9 Fig. Postnatal deletion of *Zmiz1* in LECs does not impair lymph flow.** (A) Schematic illustration of postnatal lymph flow test using BODIPY FLC16 dye. (B) Fluorescence images of P8 mesenteric lymphatic vessels in *Zmiz1*-KO pups indicate that lymph flow is not impaired after the deletion of Zmiz1, as BODIPY FLC16 dye was similarly present throughout the mesenteric lymphatic vessels of control mice (n = 3–5). Scale bars: 5 mm.
(TIF)

**S10 Fig. Ear lymphatic vasculature in *Zmiz1* deficient mice is unaltered.** (A) Wholemount immunostaining of VEGFR3 (green) in *Zmiz1* wild type and mutant postnatal day (P) 8 ears. No obvious differences in the lymphatic organization, density and valves were noted. Scale bars: 1000 μm. Three control and knockout mice each were examined.
(TIF)

**S11 Fig. Embryonic deletion of *Zmiz1* in LECs does not lead to edema or changes in early PROX1 expression.** (A) Tamoxifen schedule used for embryonic deletion of Zmiz1. Tmx, tamoxifen. (B) E14.5 control and *Zmiz1*-KO embryos. No edema was observed at E14.5 (n = 3–5; scale bars: 2 mm). (C) E14.5 control and *Zmiz1*-KO mesentery immunolabeled for PROX1 (green) and VEGFR3 (red). No differences in PROX1 and VEGFR3 expression or lymphatic vessel morphology were noted (n = 3; scale bars: 1 mm).
(TIF)

**S12 Fig. Western blot raw images.**
(TIF)

**S1 Table. List of qPCR primer sequence.**
(TIF)

## Acknowledgments

We thank Dr. Jovanny Zabaleta and Dr. Jone Garai at The Louisiana Cancer Research Consortium (LCRC) Translational Genomics Core Center for their continued support with our RNA-Seq and ATAC-Seq experiments.

## Author Contributions

**Conceptualization:** Rajan K. C., Nehal R. Patel, Maria J. Galazo, Stryder M. Meadows.

**Data curation:** Rajan K. C., Anoushka Shenoy.

**Formal analysis:** Rajan K. C.

**Funding acquisition:** Stryder M. Meadows.

**Investigation:** Rajan K. C., Nehal R. Patel, Anoushka Shenoy, Maria J. Galazo, Stryder M. Meadows.

**Methodology:** Rajan K. C., Nehal R. Patel.

**Project administration:** Stryder M. Meadows.

**Resources:** Joshua P. Scallan, Mark Y. Chiang, Maria J. Galazo, Stryder M. Meadows.

**Supervision:** Maria J. Galazo, Stryder M. Meadows.

**Validation:** Rajan K. C., Stryder M. Meadows.

**Visualization:** Rajan K. C.

**Writing – original draft:** Rajan K. C., Maria J. Galazo, Stryder M. Meadows.

**Writing – review & editing:** Rajan K. C., Nehal R. Patel, Anoushka Shenoy, Joshua P. Scallan, Mark Y. Chiang, Maria J. Galazo, Stryder M. Meadows.

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
