## [Decision Letter · Decision Letter 0]

17 Nov 2023

PONE-D-23-27443Zmiz1 is a novel regulator of lymphatic endothelial cell gene expression and functionPLOS ONE

Dear Dr. Meadows,

Thank you for submitting your manuscript to PLOS ONE. After careful consideration, we feel that it has merit but does not fully meet PLOS ONE’s publication criteria as it currently stands. Therefore, we invite you to submit a revised version of the manuscript that addresses the points raised during the review process.

We look forward to receiving your revised manuscript.

Kind regards,

Kenji Tanigaki, Ph.D., M.D.

Academic Editor

PLOS ONE

Reviewers' comments:

Reviewer's Responses to Questions

**Comments to the Author**

1. Is the manuscript technically sound, and do the data support the conclusions?

Reviewer #1: Yes

Reviewer #2: Partly

2. Has the statistical analysis been performed appropriately and rigorously? 

Reviewer #1: Yes

Reviewer #2: Yes

3. Have the authors made all data underlying the findings in their manuscript fully available?

Reviewer #1: Yes

Reviewer #2: Yes

4. Is the manuscript presented in an intelligible fashion and written in standard English?

Reviewer #1: Yes

Reviewer #2: Yes

---

## [Author Response · Author response to Decision Letter 0]

6 Dec 2023

1. Is the manuscript technically sound, and do the data support the conclusions?

Reviewer #1: Yes

Reviewer #2: Partly

We appreciate the reviewers’ comments. While our manuscript was in review, we felt we could improve the rigor of the experiments related to the ATAC-seq represented in Fig 3. In the original submission, we identified two open areas of chromatin in the Prox1 gene of HDLECs treated with control siRNA (Fig 3H). One region was previously identified as an enhancer for Prox1 expression. However, upon Zmiz1 siRNA treatment, the accessibility of the open chromatin at these two sites was markedly reduced suggesting that loss of Zmiz1 was responsible for the changes in open chromatin status. Therefore, to further assess Zmiz1 regulation of Prox1 at these sites, we conducted luciferase reporter assays. We cloned the Prox1 peaks identified from the ATAC-seq experiments into luciferase reporter vectors and performed the assay in HEK-293T cells. We note that lymphatic ECs, including HDLECs, were not used because they are notoriously difficult to transfect and HEK-293T cells are routinely used as an alternative. We observed that upon Zmiz1 siRNA treatment, luciferase activity was significantly reduced in comparison to control siRNA treated cells indicating that Zmiz1 regulates Prox1 expression at these two identified regions of DNA. Subsequently, we decided to include this experiment into the manuscript because we felt it further supported the ATAC-seq results and overall findings of the study. 

2. Has the statistical analysis been performed appropriately and rigorously?

Reviewer #1: Yes

Reviewer #2: Yes

No comment

3. Have the authors made all data underlying the findings in their manuscript fully available?

Reviewer #1: Yes

Reviewer #2: Yes

All data incorporated in this manuscript is fully available. As mentioned in the manuscript, RNA-seq and ATAC-seq data have been deposited in the Gene Expression Omnibus (GEO) database with accession no. GSE225130 and GSE225057 respectively.

4. Is the manuscript presented in an intelligible fashion and written in standard English?

Reviewer #1: Yes

Reviewer #2: Yes

No comment

---

## [Editor Report · Decision Letter 1]

28 Dec 2023

PONE-D-23-27443R1Zmiz1 is a novel regulator of lymphatic endothelial cell gene expression and functionPLOS ONE

Dear Dr. Meadows,

Thank you for submitting your manuscript to PLOS ONE. After careful consideration, we feel that it has merit but does not fully meet PLOS ONE’s publication criteria as it currently stands. Therefore, we invite you to submit a revised version of the manuscript that addresses the points raised during the review process.

We look forward to receiving your revised manuscript.

Kind regards,

Kenji Tanigaki, Ph.D., M.D.

Academic Editor

PLOS ONE

Additional Editor Comments:

We appreciate your efforts in revising your manuscript for PlosOne.

However, we regret to inform you that there was an error in the previous decision letter and some of the reviewer comments were omitted. This was due to a technical glitch in our system and we sincerely apologize for this mistake.

We have attached the complete set of reviewer comments to this letter and we kindly request you to revise your manuscript accordingly. Please also provide a point-by-point response to the comments and highlight the changes you have made in the manuscript.

We hope that you understand the situation and cooperate with us to ensure the quality of your manuscript. Please resubmit your revised manuscript and response letter within two weeks.

Thank you for your patience and understanding.

-------

Reviewer1:

Zmiz1 is a novel regulator of lymphatic endothelial cell gene expression and function

This manuscript by Rajan et al; about a novel transcriptional regulator of Prox-1 is interesting. The data is sound, and experiments were planned in a logical fashion. Data presentation and manuscript are well written and easy to follow. Invitro expts. were conducted thoroughly and results were inferred nicely. Despite a strong down regulation of Prox-1 and other lymphatic genes it is surprising that the inducible TG mice didn’t show any major lymphatic phenotype except decreased valve density. I have couple of suggestions in this regard as mentioned below.

1. It is surprising why the authors didn’t pursue any expts. with adult mice? What is the lymphatic structure and function in adult mice in cre activated Zmiz1 mice? What is the status of valves in skin?

2. With significantly decreased LEC proliferation and migration may be Zimz1 mice might have some abnormalities in lymphatic structure and function in inflammation, infection wound healing etc. it is worth trying at least one model of adult lymphatic vascular remodeling expt. With these mice.

3. Have you maintained these mice beyond P8 and observed their weight gain and lipid absorption after 3-4 wks of age.

4. In figure 4 decreased valve density is obvious. I also notice reduced lymphatic diameter in Zmiz1 mice.

5. I strongly recommend some more invivo analysis of these mice related to lymphatic density, diameter of skin and gut lymphatics. With such a significant reduction in valve density these mice should have pumping defects in skin lymphatic vessels.

Reviewer2:

The author investigated the role of Zmiz1 in lymphatic endothelial cells (LECs).

The finding that it is a regulator of the expression of Prox1, which is critically important in LECs, presents interesting data. The finding of the effect on lymphatic valve formation using the knockout mice is of some significance.

However, the presentation of the data is not sufficient, and explanations are lacking in some points.

The following points need to be improved.

Major points: additional experiments are required.

Sumoylation of Prox1 controls its ability to induce VEGFR3 expression and lymphatic phenotypes in endothelial cells (J Cell Sci　2009 )

The article above suggest that PIAS is involved in SUMOylating Prox1.

Can overexpression of Zmiz1 induce SUMOylaton of Prox1 and affect Prox1 function by e.g. Prox1 protein stability or Prox1 nuclear accumulation?

Although described as ‘similar to other cell types such as Hela,’

since there is no data for HeLa, the author needs to show immunostaining for comparison, along with BEC types such as HUVEC, which would be valuable.

Also, since the specificity of the antibodies was not shown, compare with Zmiz1 -knocked down samples.

Fig. 1G is quite important data, since the expression of genes with important functions in lymphatic vessels other than Prox1 is shown to be decreased. qRT-PCR at the mRNA level should show the decreased expression.

Also, in siRNA experiments, it is necessary to use two or more siRNAs because it is necessary to resolve the suspicion of off-target effects,

The author should show using qRT-PCR that the gene changes seen in Fig,1G (Prox1, FLT4, NR2F2, Foxc2, EFNB2, NRP2) are commonly seen as effects of at least two different siRNAs.

Since cell proliferation is decreased by siRNA, it is not easy to conclude that the results of the wound healing assay are due to decreased cell migration.

The author should provide data by using an alternative method.

In Fig. S5, it is necessary to use a sample in which the expression of Zmiz1 is sufficiently decreased. At the same time, it is necessary to show that the expression of FLT4 is also downregulated by reprobing method in Western blot.

Unless these are shown, the conclusion that Akt and Erk signaling is not affected cannot be drawn. Importantly, non-cropped figures should also be shown as supplementary figures.

In Fig. 1G and 2B, the expression of genes that are considered to be important in LECs is decreased by siRNA.　To validate these data, the author should show data on the expression of these genes by qRT-PCR.

The author showed that when Zmiz1 expression goes down, Prox1 expression goes down,

In this case, it is possible that apoptosis occurs in LECs.

Also, it is possible that tube like formation of LEC is suppressed, the author should perform the assay shown as in Fig. S4 in the article; Miyazaki et al., Cancer Sci 2014, in order to show whether the function of LECs is suppressed.

Although in Fig. 4, valve formation is decreased as a result of reduced Prox1 function, the reduced expression of FoxC2 is also expected based on the HDLEC results.

The author should show the expression of Foxc2 in the valve.

Since the valve formation phase is in the late stage of lymphatic differentiation, the author should analyze if the early lymphatic differentiation is affected,

For example, fetal skin or whole embryo at E15.5 can be immunostained for lymphatic marker proteins.

Minor points: The description needs to be changed.

The following statement in the Introduction

‘Zmiz1 function is critical in diverse developmental processes, such as angiogenesis, as demonstrated by the occurrence of vascular defects and embryonic lethality upon global deletion of Zmiz1(9).’

The author should describe in more detail as this is considered an important result in vascular biology.

The author should explain what type of ECs, EC1 and EC2 are in Fig. 1B, since it is not clear what type of ECs they are.

In Fig.S6, the author did not explain the role of Vash1 in lymphatic vessels, potential reader will not understand and the data will be meaningless. What is the role of Vash1? Also show a reference.

Similarly, the role of TMEM204 in fig. 1G needs to be described.

In discussion section, the following was described.

‘These genes induce molecular changes that stimulate LEC migration and proliferation.’ This fact is not proven in this paper, so the author should refer to and describe the results in other articles.

As to the sentence; ‘We also observed a reduction in heart valve development genes such as Dll4, Bmp2, Hey1, Snai1, and TGFβ1.’, the author should show as reference articles that demonstrated that these genes are involved in lymphatic valve formation.

The sentence would be nice if these genes were involved in both heart valve and lymphatic valve formation otherwise there is no point in listing them here.

---

## [Author Response · Author response to Decision Letter 1]

2 Feb 2024

PLEASE SEE THE ATTACHED RESPONSE TO REVIEWERS WORD DOCUMENT—IT HAS ADDITIONAL IMAGES THAT ARE NOT PART OF THE PUBLISHED WORK BUT ADDRESS REVIEWER CONCERNS/SUGGESTIONS

Additional data added but not requested by the reviewers:

In reviewing our own manuscript, we felt we could improve the rigor of the experiments related to the ATAC-seq represented in Fig 3. In the original submission, we identified two open areas of chromatin in the Prox1 gene of HDLECs treated with control siRNA (Fig 3H)—one region was previously identified as an enhancer for Prox1 expression. However, upon Zmiz1 siRNA treatment, the accessibility of the open chromatin at these two sites was markedly reduced suggesting that loss of Zmiz1 was responsible for the changes in open chromatin status. Therefore, to further assess Zmiz1 regulation of Prox1 at these sites, we conducted luciferase reporter assays. We cloned the two Prox1 peaks identified from the ATAC-seq experiments into luciferase reporter vectors and performed the assays in HEK-293T cells. We note that lymphatic ECs, including HDLECs, were not used because they are notoriously difficult to transfect and HEK-293T cells are routinely used as an alternative. We observed that upon Zmiz1 siRNA treatment, luciferase activity was significantly reduced in comparison to control siRNA treated cells further supporting that Zmiz1 regulates Prox1 expression at these two identified regions of DNA. We have included this experiment into Fig 3I, J of the manuscript because we felt it further supported the ATAC-seq results and overall findings of the study. In addition, and as Supplemental Figure 6, we showed that ZMIZ1 is expressed in the nucleus of HEK-293T cells via immunostaining and that the ZMIZ1-siRNA lead to a significant reduction in both ZMIZ1 and PROX1 mRNA compared to control siRNA in HEK-293T cells. 

Reviewer1:

Zmiz1 is a novel regulator of lymphatic endothelial cell gene expression and function

This manuscript by Rajan et al; about a novel transcriptional regulator of Prox-1 is interesting. The data is sound, and experiments were planned in a logical fashion. Data presentation and manuscript are well written and easy to follow. Invitro expts. were conducted thoroughly and results were inferred nicely. Despite a strong down regulation of Prox-1 and other lymphatic genes it is surprising that the inducible TG mice didn’t show any major lymphatic phenotype except decreased valve density. I have couple of suggestions in this regard as mentioned below.

1. It is surprising why the authors didn’t pursue any expts. with adult mice? What is the lymphatic structure and function in adult mice in cre activated Zmiz1 mice? What is the status of valves in skin?

We completely understand the reviewer’s point of view regarding experiments associated with adult mice. We were also surprised to find relatively minor lymphatic defects in the neonate Zmiz1-KO mice (addressed in more detail further down) and felt adult studies would be interesting and important to understanding the overall role of Zmiz1 in lymphatic blood vessel structure and function. With that said, we stuck to in vitro and transcriptomic methodologies, and in vivo assessment in early neonate mice because my lab is familiar with these approaches and stages, and because our experience with the lymphatic system is limited. In truth, this work is our introduction into the lymphatic vascular field and we seldom perform any experiments using adult mice. To carry out many of the adult experiments suggested by the reviewer would require a more trial and error approach and expansion of tool sets, which would be difficult to achieve in the given revision time frame. By no means are we trying to minimize the validity of the reviewer’s comments and suggestions, as we also think they are important. We are merely trying to be practical in the self-assessment of our capabilities and time constraint imposed by doing adult studies. Additionally, we feel these suggested experiments wouldn’t necessarily change the overlying implication of the paper that Zmiz1 is a regulator of lymphatic endothelial gene expression, especially Prox1, and note the reviewer reported no issues with the original experiments conducted. We certainly aim to increase our knowledge base and technical skill sets with lymphatic system-based studies going forward, but at the same time would submit that if we had performed adult studies and seen consequences in lymphatic functions, we would have been inclined to send this work to a higher impact journal. Ultimately, we feel adult work could be a separate project to further assess Zmiz1’s role in mesenteric lymphatics, ear skin lymphatics, dorsal skin lymphatics, heart lymphatics, and perhaps by performing more complex and challenging lymphatic valve pressure myography assays. We hope the reviewer understands our point of view and the subsequent responses to their additional comments/suggestions.

In reference to the reviewer’s first suggestions, we did perform an additional analysis aimed at the skin lymphatics in neonate mice. We were able to analyze the lymphatics in the ear skin via VEGFR3 immunostaining of Zmiz1 control and mutant P8 pups. Upon examination, we did not detect any clear changes in lymphatic vessel organization, density, diameters, or valve density. This result indicated that Zmiz1 is not required for lymphatic development in the early postnatal ear skin and has been added to the manuscript text and as Supporting Figure 10. Additionally, and in relationship to the nature of the reviewer’s comments, we analyzed the mesenteric lymphatics of Zmiz1 control and mutant pups at P20. In this case, we observed distinct alterations in valve density and organization, similar to P8 Zmiz1 mutant mice. Though not explicitly addressing the adult angle, we at least provide some data demonstrating that the mesentery lymphatic valve defects persist in slightly older Zmiz1-KO mice. This data has been referenced in the paper and included as Supporting Figure 8. The corresponding images are also shown below:

Regarding the absence of major lymphatic defects despite the downregulation of Prox1 and other major lymphatic players, we offer some possible explanations. Surveying the upregulated genes in Zmiz1-KD HDLECs, we found an enrichment of multiple pro-lymphangiogenic genes (Supporting Figure 3). Genes such as TGFb and Klf4 were among the upregulated genes. Increased expression of these genes, at least in part, may explain the lack of major lymphatic defects. TGF-β signaling maintains the structure of lymphatic vessels and lymphatic homeostasis such that TGF-β KO result in abnormal lymphatic vessel structure, increased lymphatic vessels network, decreased lymphatic drainage, and potentiate lymph angiogenesis (Fukasawa et al., 2021; Itoh and Watabe, 2022). Klf4 is well known for its regulation of angiogenesis. In lymphatics, Klf4 deletion resulted in defective lymphatic vessels, branching morphogenesis and decreased lymphatic density (Choi et al., 2017) and is a valve forming gene (Scallan et al., 2021). Furthermore, valve formation is a critical step during lymphatic vessel development and multiple genes, including PROX1, GATA2, FOXC2, and FOXO1 play important roles. Intriguingly, FOXO1 is significantly downregulated in Zmiz1-KD HDLECs. FOXO1 is a repressor of lymphatic mesenteric valves and FOXO1-KO mice exhibit increased mesenteric lymphatic vessels and valve numbers (Niimi et al., 2021; Scallan et al., 2021). Perhaps increased FOXO1 levels are counterbalancing the lymphatic valve defects induced in Zmiz1-KO mice. These potential explanations have been added to the discussion, which originally included references to i) Zmiz2 compensating in some manner for the loss of Zmiz1; based on our survey of single cell sequencing databases, Zmiz2 is also expressed in the various subtypes of lymphatic ECs and ii) potential differences in the in vivo and in vitro settings. 

2. With significantly decreased LEC proliferation and migration may be Zimz1 mice might have some abnormalities in lymphatic structure and function in inflammation, infection wound healing etc. it is worth trying at least one model of adult lymphatic vascular remodeling expt. With these mice.

We appreciate the reviewer’s comment and agree that it would be interesting to know whether loss of Zmiz1 and associated decreases in LEC proliferation and migration leads to changes in lymphatic structure and function in inflammation, infection wound healing, etc. However, we feel these experiments, though worthy and informative, are i) not trivial in nature, especially considering our lack of familiarity with them; ii) are somewhat beyond the scope of our manuscript linking Zmiz1 to transcriptional regulation of lymphatic gene expression; and iii) would be best addressed by future experiments aimed at further characterizing lymphatic structure and function in Zmiz1 deficient mice. 

We would also like to note that, for future direction, it is worth looking at the role of Zmiz1 in adulty mice and perform in depth in vivo analysis from looking at lymphatic density, vessel coverage, gut lymphatics, inflammations, wound healing, and infections.

3. Have you maintained these mice beyond P8 and observed their weight gain and lipid absorption after 3-4 wks of age.

We have let a few tamoxifen induced mice go for up to 3 weeks, but we did not observe any noticeable weight gain or other obvious problems typically related to dramatic lymphatic defects. As referenced in #1, we did find that similar to P8, lymphatic valves are present in fewer numbers in Zmiz1-KO mice at P20 (new S8 Fig). However, we did not perform additional BODIPY FL C16 tests at 3-4 wks of age because we felt this might not be necessary considering the same valve phenotypes at P8 were also present at P20, and we didn’t see defects in absorption of BODIPY FL C16 and subsequent lymphatic flow at P8. In terms of other methodologies to assess lipid absorption, they were not performed but would be interesting to see in future studies. 

4. In figure 4 decreased valve density is obvious. I also notice reduced lymphatic diameter in Zmiz1 mice.

We appreciate the reviewer’s keen observations on the differences in lymphatic diameters shown between Zmiz1 control and mutant mesentery in Fig 4. Although the specific Zmiz1-KO mutant shown indicates reduced vessel diameter compared to controls, this was not a consistent phenotype. We initially thought this might be the case, but as we assessed more experimental samples, no significant differences in vessel diameter were observed. Since the images shown could inadvertently suggest a change in diameter to the readers, we made a point in the figure legend to note that differences in diameter were not observed between Zmiz1 control and mutant vessels. We also include images below for the author highlighting the similar vessel diameters between Zmiz1 control and mutant mesentery. 

5. I strongly recommend some more in vivo analysis of these mice related to lymphatic density, diameter of skin and gut lymphatics. With such a significant reduction in valve density these mice should have pumping defects in skin lymphatic vessels.

We appreciate Reviewer 1 for commenting on the need for more in vivo experiments. As noted in our response to comment #1, we analyzed the ear skin lymphatics at P8 and saw no gross discernable differences in valve morphology or lymphatic vessel density, diameter, and organization (S10 Fig). We also noted in comment #4 that mesenteric lymphatic vessel calibers did not appear significantly different between control and Zmiz1-KO mice (P8 and P20). This is referenced in the Figure 4 legend. For these reasons, we did not provide further in-depth in vivo experiments but cannot rule out the possibility that there are pumping defects in the skin lymphatics or in adults. As mentioned before, we fully believe these types of in-depth in vivo analyses are valid and critical to future directions aimed at further elucidating the role of Zmiz1 in lymphatic structure and function, especially in adults. 

Reviewer2:

The author investigated the role of Zmiz1 in lymphatic endothelial cells (LECs).

The finding that it is a regulator of the expression of Prox1, which is critically important in LECs, presents interesting data. The finding of the effect on lymphatic valve formation using the knockout mice is of some significance.

However, the presentation of the data is not sufficient, and explanations are lacking in some points.

The following points need to be improved.

Major points: additional experiments are required.

1. Sumoylation of Prox1 controls its ability to induce VEGFR3 expression and lymphatic phenotypes in endothelial cells (J Cell Sci, 2009). The article above suggests that PIAS is involved in SUMOylating Prox1. Can overexpression of Zmiz1 induce SUMOylaton of Prox1 and affect Prox1 function by e.g. Prox1 protein stability or Prox1 nuclear accumulation?

As mentioned by Reviewer 2, studies have shown Prox1 SUMOylation effecting vegfr3 expression. Zmiz1 is a member of PIAS proteins based on structural domains, however PIAS activity of Zmiz1 has not been assessed extensively, and SUMOylation activity of Zmiz1 has only been associated with the androgen receptor (Sharma et al., 2003). In this study, we focused on the transcriptional role of Zmiz1 in regulating lymphatic endothelial cell gene expression, including Prox1. The SUMOylation aspect of Zmiz1 is an interesting point, as it is unknown if Zmiz1 might regulate Prox1 function via SUMOylation. However, though interesting and worthy of future investigation, we feel that the potential role of Zmiz1 in Prox1 SUMOylation and its subsequent function go beyond the scope of this study, and therefore, we did not pursue this line of investigation.

2. Although described as ‘similar to other cell types such as Hela,’ since there is no data for HeLa, the author needs to show immunostaining for comparison, along with BEC types such as HUVEC, which would be valuable.

Based upon the nature of the reviewer’s request, instead of solely referencing the study with Hela cells, we now reference proteinatlas.org (Human protein atlas database: https://www.proteinatlas.org/ENSG00000108175-ZMIZ1/single+cell+type) and other various publications indicating expression of Zmiz1 in multiple cell types and in the nucleus. In addition, we can confirm that Zmiz1 is expressed in HUVECs, as well as other endothelial cell lines (human teloHAEC and mouse MS1) and in the endothelial cells of mice (see below figure). However, this data is being used for a manuscript in preparation detailing Zmiz1’s role in angiogenesis and therefore we have not included it in this paper. We hope Reviewer 2 understands our reasoning. I have included some of the data in good faith showing that Zmiz1 is expressed in endothelial cells from different databases, as well as in the retinal blood vessels of mice, as revealed in our studies:

Summary of EC-specific Zmiz1 expression within various tissues. A, Analysis of Zmiz1 expression levels in ECs and total tissue of P7 brain, kidney, liver, lung, and adult brain using the Vascular Endothelial Cell Trans-Omics Resource Database (VECTRDB). B, Zmiz1 expression in each cell type isolated from an adult mouse brain can be visualized within the violin plots. C, Analysis of Zmiz1 expression levels in P7 isolated brain ECs and its subtypes using single cell RNA-seq data in VECTRDB. D, Representative immunofluorescent images of P7 retinas stained with IB4 and Zmiz1 (A - artery and V - vein). Scale bars: 200 μm. E, Expression levels of Zmiz1 in murine retinal endothelial cells during postnatal development determined using the available bulk RNA sequencing data.

We also note that we tried numerous anti-Zmiz1 antibodies but were never able to detect Zmiz1 in the different endothelial cell lines via western blot or immunofluorescent staining. Rather, we had to use qPCR to show reduced Zmiz1 expression in the cell lines subjected to siRNA targeting Zmiz1. However, and for the reviewer’s reference only, we showed that an HA-tagged version of ZMIZ1 could be detected in the nucleus of mouse MS1 cells:

3. Also, since the specificity of the antibodies was not shown, compare with Z

---

## [Decision Letter · Decision Letter 2]

22 Feb 2024

PONE-D-23-27443R2Zmiz1 is a novel regulator of lymphatic endothelial cell gene expression and functionPLOS ONE

Dear Dr. Meadows,

Thank you for submitting your manuscript to PLOS ONE. After careful consideration, we feel that it has merit but does not fully meet PLOS ONE’s publication criteria as it currently stands. Therefore, we invite you to submit a revised version of the manuscript that addresses the points raised during the review process.

We look forward to receiving your revised manuscript.

Kind regards,

Kenji Tanigaki, Ph.D., M.D.

Academic Editor

PLOS ONE

Reviewers' comments:

Reviewer's Responses to Questions

**Comments to the Author**

1. If the authors have adequately addressed your comments raised in a previous round of review and you feel that this manuscript is now acceptable for publication, you may indicate that here to bypass the “Comments to the Author” section, enter your conflict of interest statement in the “Confidential to Editor” section, and submit your "Accept" recommendation.

Reviewer #2: (No Response)

2. Is the manuscript technically sound, and do the data support the conclusions?

Reviewer #2: Partly

3. Has the statistical analysis been performed appropriately and rigorously? 

Reviewer #2: Yes

4. Have the authors made all data underlying the findings in their manuscript fully available?

Reviewer #2: Yes

5. Is the manuscript presented in an intelligible fashion and written in standard English?

Reviewer #2: Yes

6. Review Comments to the Author

Reviewer #2: The authors have experimented and responded sincerely to most of the comments,

However, I believe the following two points are still critically important.

1. Not examining Sumolylation of Prox1 is not an appropriate attitude of an author, as it lacks respect for the work of past researchers who examined Sumoylation of Prox1.

They did not even include a reference and to not discuss the issue.

It is not natural to simply not examine the effect of Prox1 Sumolylation because this is not a difficult experiment.

2. Even though there are indivisual siRNAs, the authors did not examine the off-target effect of siRNAs,

it is not appropriate response because it may downgrade the scientific quality of the data.

Minor point:

Listing the references in No. 54 and 55 is not balanced in context. The original articles, not the review article should be added as reference articles.

7. PLOS authors have the option to publish the peer review history of their article (what does this mean?). If published, this will include your full peer review and any attached files.

Reviewer #2: No

---

## [Author Response · Author response to Decision Letter 2]

22 Mar 2024

PLEASE SEE ATTACHED RESPONSE TO REVIEWER DOCUMENT

Reviewer #2: The authors have experimented and responded sincerely to most of the comments. However, I believe the following two points are still critically important.

1. Not examining Sumolylation of Prox1 is not an appropriate attitude of an author, as it lacks respect for the work of past researchers who examined Sumoylation of Prox1. They did not even include a reference and to not discuss the issue. It is not natural to simply not examine the effect of Prox1 Sumolylation because this is not a difficult experiment.

With guidance from the Editor, and specifically in respect to lymphatic ECs, we performed the following assay to address Reviewer #2’s demand. We treated HDLECs with control and Zmiz1 siRNAs (48 hours) and assessed PROX1 SUMOylation by immunoprecipitating PROX1 and then immunoblotting for both PROX1 and SUMO1. Based on the results (shown below), and as best as we can tell, we did not see any changes in PROX1 SUMOylation (performed in triplicate; one blot shown). We note that as expected and previously shown in our studies, PROX1 levels were markedly reduced in Zmiz1 siRNA treated HDLECs compared to controls. Immunostaining for SUMO1 indicated protein detection at the same size that PROX1 was immuno-detected in both control and Zmiz1 siRNA HDLECs. This might suggest detection of a SUMOylated form of PROX1, though no obvious changes in typical PROX1 size or additional larger bands were observed as in the Pan et al., studies (denaturing and non-denaturing conditions showed similar results in our experiments). However, we note that the Pan et al., studies used non-lymphatic cells and the one vascular cell line they used was a fusion of HUVEC to a human lung cancer cell. Thus, it is unclear if PROX1 SUMOylation takes place in lymphatic cells, though it clearly does in other cell types. 

Densitometric measurements of PROX1/SUMO1 ratios in control and Zmiz1 siRNA treated HDLECs showed no statistical differences in expression levels. Thus, based on the lack of an additional, larger size PROX1 band and no changes in SUMO1 expression levels, our experiments indicate loss of Zmiz1 did not have an effect on PROX1 SUMOylation, at least via SUMO1. Since the reviewer was adamant about performing PROX1 SUMOylation experiments, we have included these findings as S7 Fig with additional text being incorporated into the results section and S7 Fig legend. However, we note that we are not confident that these results show or do not show Zmiz1 mediated PROX1 SUMOylation in lymphatic ECs.

We felt it most important and relevant to conduct the experiments in HDLECs, as opposed to non-lymphatic EC types, which were used in the reference provided by the reviewer. However, to further assess possible SUMOylation of PROX1 by ZMIZ1, we performed an additional experiment in human 293T cells, similar to the original suggestion of the reviewer and similar to experiments in the reference provided by the reviewer. Epitope tagged versions of ZMIZ1 (Zmiz1-FLAG; cloned in our lab) and SUMO1-HA (Addgene, 21154) were transfected independently and together in 293T cells for 48 hours. Proteins were then subjected to western blot analysis (shown below). Immunoblotting for HA in 293T cells transfected with SUMO1-HA alone showed a strong band at approximately 95 kDa, which also coincided with the size for PROX1 seen in all samples. This result was similar to the experiment above and may indicate that PROX1 was SUMOylated by SUMO1. On the other hand, staining for PROX1 itself only resulted in a single band at ~95 kDa and not a second larger band that might be expected if some of PROX1 was SUMOylated. Zmiz1-FLAG transfected 293T cells displayed a band at the expected 115 kDa size (arrow), though this band was moderate in signal strength and directly above a non-specific band. Importantly, transfection of constructs for both epitope-tagged proteins revealed no obvious changes in size of PROX1 or additional larger PROX1 bands as compared to the control or single transfected lanes. Furthermore, HA immunostaining showed the same size band (~95 kDa) as in the SUMO1-HA individually transfected cells, while ZMIZ1 was also detected. However, we noted that expression of these proteins, including PROX1 appeared reduced proportionally; we don’t have an explanation for this aspect of the experiment. Ultimately, these results may indicate PROX1 SUMOylation was unaltered in the presence of excess ZMIZ1, though admittedly, it’s difficult to interpret. For this reason, we have shown the images of the experiment to the reviewer below but have not included it into the paper. We hope this demonstrates to the reviewer that our experiments were conducted in good faith but because of the results we were reluctant to include them into the final paper. Moreover, and as indicated with the previous experiment, we are also hesitant to include the above experiment (PROX1 IP in HDLECs) in the paper, but ultimately did because of the reviewer’s request. Though it is our opinion, that these experiments: a) were not as easy to conduct/interpret as was indicated by the reviewer, b) may not sufficiently address the reviewer’s request c) still do not alter the main thrust of the paper that Zmiz1 transcriptionally regulates lymphatic gene expression, including Prox1 and d) might be better left off entirely from the manuscript, as we don’t feel they are conclusive, though we have tried diligently to assess SUMOylation of Prox1 via Zmiz1 and further investigation would take much longer than we feel is warranted considering our efforts into the manuscript studies.

2. Even though there are indivisual siRNAs, the authors did not examine the off-target effect of siRNAs, it is not appropriate response because it may downgrade the scientific quality of the data.

As noted previously, Zmiz1 siRNA were obtained as a SMARTpool ON-TARGETplus siRNA from Horizon Discovery, which is a mixture of 4 siRNAs provided as a single reagent—providing advantages in both potency and specificity. To address the reviewer’s concern about potential off-target effects, we performed siRNA transfections in HDLECs with 2 of the 4 Zmiz1 siRNAs individually and compared them to control siRNA treated cells—2 independent Zmiz1 siRNAs were tested based on the guidance of the Editor. After 48 hours of treatment, RNAs were isolated and qPCR analysis was conducted to assess mRNA levels of ZMIZ1, PROX1, EFNB2, FLT, COUPTFII, and FOXC2. All mRNAs assessed were substantially reduced, similar to previous results utilizing the Zmiz1 SMARTpool siRNAs. Thus, the results indicate high specificity of the Zmiz1 siRNAs and subsequently suggest limited off-target effects. The results of these experiments are noted in the methods section and the data is presented as an additional part of S5 Fig. We believe these experiments address the reviewer’s concern over potential Zmiz1 siRNA off-target effects. 

Minor point:

Listing the references in No. 54 and 55 is not balanced in context. The original articles, not the review article should be added as reference articles

This was an error on our part. We thank the reviewer for catching this and have subsequently rebalanced the references and used the original articles in citation #56 and #57.

---

## [Decision Letter · Decision Letter 3]

9 Apr 2024

PONE-D-23-27443R3Zmiz1 is a novel regulator of lymphatic endothelial cell gene expression and functionPLOS ONE

Dear Dr. Meadows,

Thank you for submitting your manuscript to PLOS ONE. After careful consideration, we feel that it has merit but does not fully meet PLOS ONE’s publication criteria as it currently stands. Therefore, we invite you to submit a revised version of the manuscript that addresses the points raised during the review process

After a thorough review of your manuscript and considering the insightful comments provided by our reviewer, we believe that your work holds significant academic value and potential for publication in our journal.

However, before we can proceed further, we kindly ask that you undertake a minor revision of your manuscript. Specifically, we request that you revisit the data interpretation in light of the reviewer’s suggestions. We believe that these revisions will not only clarify the data but also strengthen the overall impact of your research findings.

Please find attached the detailed comments from the reviewers. We would appreciate it if you could submit the revised manuscript by May 24 2024 11:59PM. If you will need more time than this to complete your revisions, please reply to this message or contact the journal office at plosone@plos.org. We appreciate your attention to this matter and look forward to receiving your revised manuscript.

Please include the following items when submitting your revised manuscript:A rebuttal letter that responds to each point raised by the academic editor and reviewer(s). You should upload this letter as a separate file labeled 'Response to Reviewers'.A marked-up copy of your manuscript that highlights changes made to the original version. You should upload this as a separate file labeled 'Revised Manuscript with Track Changes'.An unmarked version of your revised paper without tracked changes. You should upload this as a separate file labeled 'Manuscript'.If applicable, we recommend that you deposit your laboratory protocols in protocols.io to enhance the reproducibility of your results. Protocols.io assigns your protocol its own identifier (DOI) so that it can be cited independently in the future. For instructions see: https://journals.plos.org/plosone/s/submission-guidelines#loc-laboratory-protocols. Additionally, PLOS ONE offers an option for publishing peer-reviewed Lab Protocol articles, which describe protocols hosted on protocols.io. Read more information on sharing protocols at https://plos.org/protocols?utm_medium=editorial-email&utm_source=authorletters&utm_campaign=protocols.

We look forward to receiving your revised manuscript.

Kind regards,

Kenji Tanigaki, Ph.D., M.D.

Academic Editor

PLOS ONE

Journal Requirements:

Reviewers' comments:

Reviewer's Responses to Questions

**Comments to the Author**

1. If the authors have adequately addressed your comments raised in a previous round of review and you feel that this manuscript is now acceptable for publication, you may indicate that here to bypass the “Comments to the Author” section, enter your conflict of interest statement in the “Confidential to Editor” section, and submit your "Accept" recommendation.

Reviewer #2: (No Response)

2. Is the manuscript technically sound, and do the data support the conclusions?

Reviewer #2: Partly

3. Has the statistical analysis been performed appropriately and rigorously? 

Reviewer #2: Yes

4. Have the authors made all data underlying the findings in their manuscript fully available?

Reviewer #2: Yes

5. Is the manuscript presented in an intelligible fashion and written in standard English?

Reviewer #2: Yes

6. Review Comments to the Author

Reviewer #2: Since a review is a communication between the author and the reviewer,

It sounds very rude to say 'Adamant'.

Experiments or modifications requested by the reviewer could be rejected by the authors if they can be scientifically justified or stated.

The author's scientific communication skills are disappointing, but I admit the new data presented by authors is the results of their efforts. This revision could be acceptable if the following points can be addressed.

Pan's group has shown by immunoprecipitation experiments that SUMOylation of Prox1 occurs using LEC (Fig. 1C). Therefore, the author is wrong in assuming that we do not know whether SUMOylation of Prox1 occurs in LECs. This is disappointing because it shows that the author is not capable of carefully reading the prior report.

Also, and more importantly, that the Prox1 band is shown at 95 kDa,

It is more natural to assume that Prox1 is originally estimated to be around 83-85 KDa and that most Prox1 is SUMOylated. The author misinterpreted the experimental facts by missing them. At the same time, Pan's group may have mistakenly assumed that the 95 kDa protein is deSUMOylated form of Prox1.

Therefore, most of Prox1 is considered to be SUMOylated at this point.

Then the experiment of overexpression of SUMO and Zmiz1 using 293T is not very meaningful.

Supplementary figure 7 does not show that the cell extracts used for IP had the same amount of protein, and the quantification method was wrong.

The data should originally show that the same protein amounts were used in the control siRNA and Zmiz1 siRNA samples.

More precisely, the graph in fig. S7B should be shown after showing the amount of internal control protein is equally loaded. Otherwise, it is meaningless.

This is a basic rule when performing immunoprecipitaion assay. Therefore, this experiment needs to be redone. However, since it is understood that to investigate SUMOylation of Prox1 in depth, experiments such as SUMO1 knockdown will be necessary, I do not demand the authors further it in this revision.

Considering the above, Supplementary Figure 7 and the related text can be removed.

However, please explain the discrepancy since Prox1 is shown at 85 kDa in Fig. 4 and 95 kDa in figure S7. This is a problem of the reliability of the authors’ data and ability to interpret the data.

I think they have addressed my other comments well.

7. PLOS authors have the option to publish the peer review history of their article (what does this mean?). If published, this will include your full peer review and any attached files.

Reviewer #2: No

---

## [Author Response · Author response to Decision Letter 3]

13 Apr 2024

*Please see Response to Reviewer document with images*

Reviewer #2: Since a review is a communication between the author and the reviewer,

It sounds very rude to say 'Adamant'. Experiments or modifications requested by the reviewer could be rejected by the authors if they can be scientifically justified or stated.

The author's scientific communication skills are disappointing, but I admit the new data presented by authors is the results of their efforts. This revision could be acceptable if the following points can be addressed.

Pan's group has shown by immunoprecipitation experiments that SUMOylation of Prox1 occurs using LEC (Fig. 1C). Therefore, the author is wrong in assuming that we do not know whether SUMOylation of Prox1 occurs in LECs. This is disappointing because it shows that the author is not capable of carefully reading the prior report.

Also, and more importantly, that the Prox1 band is shown at 95 kDa,

It is more natural to assume that Prox1 is originally estimated to be around 83-85 KDa and that most Prox1 is SUMOylated. The author misinterpreted the experimental facts by missing them. At the same time, Pan's group may have mistakenly assumed that the 95 kDa protein is deSUMOylated form of Prox1.

Therefore, most of Prox1 is considered to be SUMOylated at this point.

Then the experiment of overexpression of SUMO and Zmiz1 using 293T is not very meaningful.

Supplementary figure 7 does not show that the cell extracts used for IP had the same amount of protein, and the quantification method was wrong.

The data should originally show that the same protein amounts were used in the control siRNA and Zmiz1 siRNA samples.

More precisely, the graph in fig. S7B should be shown after showing the amount of internal control protein is equally loaded. Otherwise, it is meaningless.

This is a basic rule when performing immunoprecipitaion assay. Therefore, this experiment needs to be redone. However, since it is understood that to investigate SUMOylation of Prox1 in depth, experiments such as SUMO1 knockdown will be necessary, I do not demand the authors further it in this revision.

Considering the above, Supplementary Figure 7 and the related text can be removed.

We have removed S7 Fig and the related text, and subsequently reordered the supplementary figures remaining in the manuscript.

However, please explain the discrepancy since Prox1 is shown at 85 kDa in Fig. 4 and 95 kDa in figure S7. This is a problem of the reliability of the authors’ data and ability to interpret the data.

Please see raw images of Fig 4 and previously S7 Fig below. In both cases, you will see that the PROX1 bands appear to be in the same location in the blots (between the 100 kDa and 75 kDa molecular weight markers; dotted lines were added to help better see this as well). Therefore, we don’t believe there are any discrepancies between the experiments—we believe we are picking up the same sized PROX1 band. We indicated the PROX1 band at 83 kDa in Fig 4 because the antibody used from Abcam (ab199359) predicts a size of 83 kDa via western blot and is how they label it, notwithstanding a potential SUMOylated form of PROX1. Of course, it is difficult to know the exact size based on the difference between the molecular marker sizes. In S7 Fig, we noted the PROX1 band at ~95 kDa primarily based on 2 factors, though this may or may not have been correct. Pan’s group indicated PROX1 band sizes at 95 kDA and 110 kDA when SUMOylated. It was clear that our band was not 110 kDa. Since immunoblotting for SUMO1 and PROX1 in the IP experiments suggested detection of a SUMOylated form in between 100 – 75 kDa, we thought it was most likely the 95 kDA version of PROX1 and thus used this size description. However, it is difficult to know the exact size, as there might not be an easily noticeable difference between 83 and 95 kDa. In looking at both blots, there doesn’t appear to be any noticeable differences in size and thus we don’t believe there are discrepancies, but rather our error in how we perceived the sizes. We may be picking up SUMOylated PROX1 in Fig 4, but since we have removed S7 Fig and because it is difficult to compare and know the exact size of our PROX1 bands with Pan’s group (plus SUMOylated versus non-SUMOylated bands), we decided to take out any specific reference to size and instead show the 100 and 75 kDa molecular sizes as reference (see updated Fig 4a below).

---

## [Editor Report · Decision Letter 4]

16 Apr 2024

Zmiz1 is a novel regulator of lymphatic endothelial cell gene expression and function

PONE-D-23-27443R4

Dear Dr. Meadows,

We’re pleased to inform you that your manuscript has been judged scientifically suitable for publication and will be formally accepted for publication once it meets all outstanding technical requirements.

Kind regards,

Kenji Tanigaki, Ph.D., M.D.

Academic Editor

PLOS ONE
---

## [Editor Report · Acceptance letter]

26 Apr 2024

PONE-D-23-27443R4 

PLOS ONE

Dear Dr. Meadows, 

I'm pleased to inform you that your manuscript has been deemed suitable for publication in PLOS ONE. Congratulations! Your manuscript is now being handed over to our production team.

Kind regards, 

on behalf of

Dr. Kenji Tanigaki 

Academic Editor

PLOS ONE